# Topic modeling identifies novel genetic loci associated with multimorbidities in UK Biobank

## Graphical abstract

## Authors

Yidong Zhang, Xilin Jiang, Alexander J. Mentzer, Gil McVean, Gerton Lunter

## Correspondence

gil.mcvean@bdi.ox.ac.uk (G.M.), g.a.lunter@umcg.nl (G.L.)

## In brief

Zhang et al. develop and apply computational methods to identify clusters of disease multimorbidity among participants in UK Biobank. The characterization of such disease topics identifies novel genetic loci, can improve risk prediction for some disorders, and identifies unique characteristics associated with a healthy, disease-free phenotype.

## Highlights

- Topic modeling of diseases in UK Biobank identifies clusters of multimorbidity

- Priors informed by known disease relationships can improve power

- Topic-GWAS finds novel genetic associations and can improve risk prediction

- A healthy topic is associated with specific behaviors and genetic variants

Zhang et al., 2023, Cell Genomics 3, 100371
August 9, 2023 © 2023 The Authors.

# Cell Genomics

CellPress

## Article

# Topic modeling identifies novel genetic loci associated with multimorbidities in UK Biobank

Yidong Zhang,[1,2,3] Xilin Jiang,[1,4,5,6,7,8] Alexander J. Mentzer,[1,5] Gil McVean,[1,*] and Gerton Lunter[9,10,11,*]

[1]Big Data Institute, Li Ka Shing Centre for Health Information and Discovery, University of Oxford, Oxford OX3 7LF, UK
[2]Chinese Academy of Medical Sciences Oxford Institute, Nuffield Department of Medicine, University of Oxford, Oxford OX3 7BN, UK
[3]Department of Radiation Oncology, Peking Union Medical College Hospital, Chinese Academy of Medical Sciences and Peking Union Medical College, Beijing 100006, China
[4]Department of Statistics, University of Oxford, Oxford OX1 3LB, UK
[5]Wellcome Centre for Human Genetics, Nuffield Department of Medicine, University of Oxford, Oxford OX3 7BN, UK
[6]Department of Epidemiology, Harvard T.H. Chan School of Public Health, Boston, MA 02115, USA
[7]Victor Phillip Dahdaleh Heart and Lung Research Institute, University of Cambridge, Cambridge CB2 0SR, UK
[8]Heart and Lung Research Institute, University of Cambridge, Cambridge CB2 0BB, UK
[9]MRC Weatherall Institute of Molecular Medicine, John Radcliffe Hospital, University of Oxford, Oxford OX3 9DS, UK
[10]Department of Epidemiology, University Medical Center Groningen, University of Groningen, Groningen 9700 RB, the Netherlands
[11]Lead contact
*Correspondence: gil.mcvean@bdi.ox.ac.uk (G.M.), g.a.lunter@umcg.nl (G.L.)

## SUMMARY

Many diseases show patterns of co-occurrence, possibly driven by systemic dysregulation of underlying processes affecting multiple traits. We have developed a method (treeLFA) for identifying such multimorbidities from routine health-care data, which combines topic modeling with an informative prior derived from medical ontology. We apply treeLFA to UK Biobank data and identify a variety of topics representing multimorbidity clusters, including a healthy topic. We find that loci identified using topic weights as traits in a genome-wide association study (GWAS) analysis, which we validated with a range of approaches, only partially overlap with loci from GWASs on constituent single diseases. We also show that treeLFA improves upon existing methods like latent Dirichlet allocation in various ways. Overall, our findings indicate that topic models can characterize multimorbidity patterns and that genetic analysis of these patterns can provide insight into the etiology of complex traits that cannot be determined from the analysis of constituent traits alone.

## INTRODUCTION

Multimorbidity is defined as the co-existence of multiple chronic conditions. Its prevalence has increased because of a worldwide increase in life expectancy,[1–3] and it is associated with substantially lower quality of life,[3,4] worse clinical outcomes,[4] and increased health-care expenditure.[5] The management of multimorbidity is a major challenge for modern health-care systems, given that most guidelines and research are targeted at single diseases.

Several common multimorbidity patterns, such as a cluster composed of cardiovascular and mental health disorders and a musculoskeletal disease cluster, have been identified from literature reviews.[3,6] In recent years, the widespread adoption of electronic health records (EHRs) has enabled the systematic study of multimorbidity using various approaches, including factor analysis,[7,8] clustering,[8–10] graph- or network-based methods,[11–13] and statistical models such as latent class analysis.[14–17] These approaches have not only validated previous findings[15,17,18] and identified additional multimorbidity patterns,[15,19] but also enabled downstream analyses such as identifying the clinical events and outcomes associated with specific multimorbidity patterns.[14,20]

The existence of common multimorbidity patterns raises the question of their etiology. One way to approach this question is to analyze the identified multimorbidity patterns together with appropriate omics data to determine the biological pathways involved. These analyses have been made possible with the establishment of biobanks linking individuals' biological samples and genetic information to their EHR.[21–23] A recent study investigating genome-wide association studies (GWASs) of 439 common diseases recorded in UK Biobank (UKB) hospital inpatient data found that 46% of multimorbidity disease pairs have evidence of shared genetics,[12] suggesting that this may be a fruitful approach.

Intrinsic to the study of multimorbidity is the joint analysis of multiple disease phenotypes, for which various multitrait GWAS methods have been developed.[24–31] Several of these methods use dimension reduction transformations on the original traits to define new composite traits.[30,31] For instance, topic models such as latent Dirichlet allocation (LDA)[32] were originally developed to model word occurrence in text documents using topics, but can also be used to find multimorbidity clusters from diagnosis data by viewing individuals as "documents" and diseases as "words." The inferred topics are mathematical representations of diseases that tend to co-occur within the

same individual.[33] Based on LDA, advanced topic models with specialized features have been developed. For instance, mix-EHR can cope with multimodal medical data, including lab tests, diagnoses, and clinical notes.[34] In terms of downstream analysis, earlier studies have shown that joining single diseases into topics increases statistical power for genetic association study and helps to disentangle pleiotropic effects of several known genetic loci.[35–37]

Despite these advances, existing methods all have limitations. First, diagnostic data are often binary in nature, with zeros and ones representing the absence and presence of diseases, yet topic models based on LDA were designed for count data, while algorithms designed for binary data[38] have not found wide application in biomedical studies. Second, for sparse diagnosis data in biobanks, inclusion of prior domain knowledge can improve inference results.[38–44] Medical ontologies like the ICD-10 disease classification system[45] encode the complex relationships of diseases as a hierarchical structure that is amenable to mathematical analysis,[46–48] yet their use as prior is still limited. Third, model selection and hyperparameter learning have a strong impact on final results, yet principled approaches are still lacking for many methods.[9,49–54] In addition, methods not based on statistical foundations typically lack estimates of uncertainty, which makes further downstream analyses difficult.

Here, we develop and validate an analytic framework for the study of multimorbidity, built around "treeLFA" (latent factor allocation with a tree-structured prior), a statistical model to identify multimorbidity clusters of common diseases based on their co-occurrence patterns and an informed prior for topics. Applying treeLFA to Hospital Episode Statistics (HES) data extracted from UKB, we find multimorbidity clusters in the form of disease "topics." Performing GWAS on the quantitative traits defined by individuals' weights for these topics (topic-GWAS), we show that the approach identifies novel loci that correlate in expected ways with several genomic annotations. We also show that topic-GWAS can improve genetic risk prediction for multiple disorders on the test data, in particular immune disorders, and those for which currently few associated loci are known.

## RESULTS

### Overview of treeLFA

treeLFA is a topic model designed to identify multimorbidity clusters in the form of topics of diseases from binary diagnosis data. The presence and absence of S disease codes for D individuals are modeled by factoring the $D \times S$ Bernoulli probability matrix into a $K \times S$ topic matrix ($\varphi$) and a $D \times K$ topic weight matrix ($\theta$) (Figure 1A). In this way, an individual's Bernoulli probability for a disease code is a mixture of the Bernoulli probability of this code in all topics, with the mixing coefficients specified by this individual's weights for topics. The likelihood for observations can be expressed as:

$$P(W|\theta, \Phi) = \prod_{d,s} \text{Bernoulli}\left(W_{d,s} \middle| [\theta \cdot \Phi]_{d,s}\right).$$

Here, W is the input data, and $P(W_{d,s} = 1)$ is the probability of disease code s diagnosed for individual d. $\varphi$ is the topic-disease

matrix (each row a topic and each column a disease), and $\theta$ is the topic weights matrix (each row an individual and each column a topic). $P(W_{d,s} = 1)$ is specified by the corresponding entry in the product of matrices $\theta$ and $\varphi$: $[\theta \cdot \varphi]_{d,s}$.

This model differs from LDA in three ways (Figures 1B and 1C). First, LDA samples diseases (or words) according to a multinomial distribution, so that diseases can occur multiple times, while treeLFA allows only presence or absence. Second, LDA conditions on the number of observed diseases, whereas for treeLFA the number of diseases is informative. Third, treeLFA uses an informative prior on topic vectors ($\varphi_k$) guided by a tree-structured ontology such as ICD-10 (see STAR Methods). This prior has the property that diseases closely related on the tree tend to have correlated probabilities in the same topic. The main effect of the prior is that when the input data are not strong enough for the inference of stable multimorbidity clusters, the model tends to favor the structure of the disease ontology in topics, instead of inferring unstable noisy patterns.

### Validation of treeLFA and comparison with related topic models

We assessed treeLFA's performance in a simulation experiment, comparing it with the same model but without an informative tree prior (flatLFA; Figure 2A) and to LDA. We simulated eight groups of datasets (Table S1A) to test the model with respect to the degree of multimorbidity in the data ($\alpha$), the size of the data (D), and the correctness of the prior (see STAR Methods and Figures 2B, 2C, S1A, and S1B). The performance of topic models was evaluated with two metrics: $\Delta\varphi$, averaged absolute per-disease difference in probability between aligned true and inferred topics, and $R_{pl}$, ratio of averaged per-individual predictive likelihood for treeLFA and flatLFA (see STAR Methods).

On datasets simulated using the correct tree prior, treeLFA performs better than flatLFA (Figures 2D–2G and Table S1B). This is most pronounced for small datasets with strong multimorbidity (Figure 2D; $\Delta\varphi$ 0.012 ± 0.004 [treeLFA] and 0.025 ± 0.010 [flatLFA]; $R_{pl}$ 1.003 ± 0.002), while for larger datasets, the two models show similar performance (Figure 2E; $\Delta\varphi$ 0.009 ± 0.006 [treeLFA] and 0.011 ± 0.005 [flatLFA]). treeLFA outperforms LDA except for large datasets with weak multimorbidity, where both models give accurate inference. For simulations using incorrect prior, the performances of flatLFA and treeLFA are similar across the four parameter combinations (Figure S1), indicating that treeLFA is robust against prior misspecification. Overall, these results indicate that the three models give accurate results when sufficient training data are available; but when the tree prior is correct, treeLFA performs better than flatLFA and LDA, particularly when training data are limited.

### Topics of ICD-10 codes inferred from UK Biobank data

To investigate the properties of treeLFA on real-world data, we built an exploratory diagnosis dataset consisting of 100 common diseases in UKB (top 100 UKB dataset; see STAR Methods and Table S2A). We then trained treeLFA on this dataset with an initial K = 11 topics (see below for a discussion of the optimal number of topics).

The inferred topics include an "empty" topic, in which all codes have near-zero probability of occurring. Its associated entry

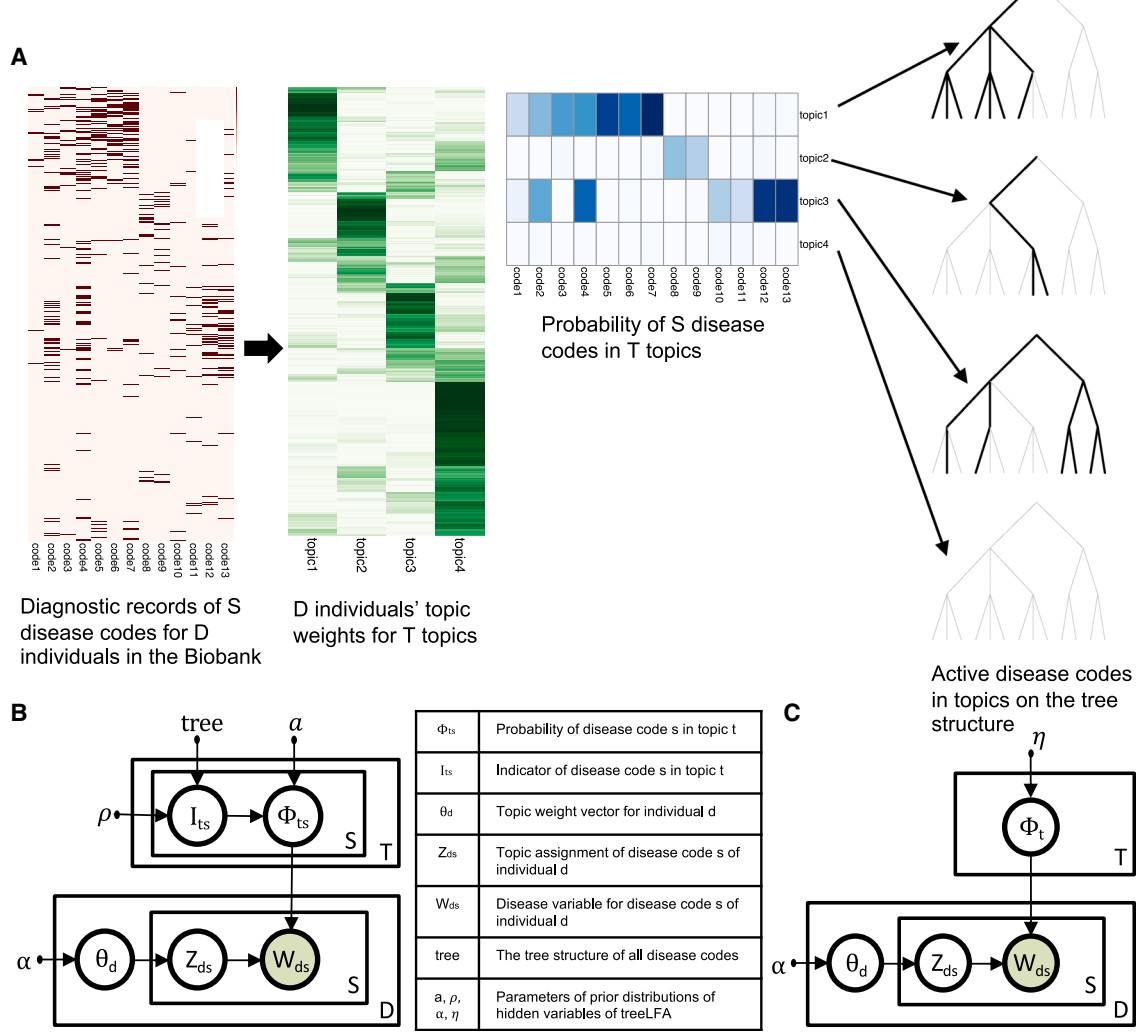

**Figure 1. Schematic for topic modeling of diagnosis data with treeLFA**

(A) The matrix of Bernoulli probability for D × S disease variables is factored into a topic matrix ($\theta$) and a topic weight matrix ($\varphi$). Each topic is composed of S Bernoulli probability for the S disease codes, and each individual has K weights for the K topics. Active disease codes in the four topics are highlighted on the tree structure of disease codes specified by a medical ontology, on which the 13 leaf nodes correspond to 13 disease codes.

(B) Graphical model of treeLFA.

(C) Graphical model of LDA for comparison with treeLFA. $\eta$ is the Dirichlet prior for topics.

See also STAR Methods.

in the optimal Dirichlet prior parameter vector ($\alpha$) is large (0.585) compared with other topics (0.016–0.06), indicating that the empty topic is frequently assigned to an individual's disease profile. The remaining topics all contain active codes. Most topics are sparse (eight topics contain fewer than 10 high-probability [>0.2] codes), but the model also infers dense topics, such as topics 8 and 10, which include 41 and 43 high-probability codes, respectively (Figure 3A and Table S2B). To assess whether codes tend to be specific to a topic, we scaled their probabilities across topics to make the largest probability 1. We found that codes are typically specific to topics: most (87/100) are active (scaled probability >0.5) in three or fewer topics. However, some codes are active in many topics, such as I10 (essential hy-

pertension, active in six topics) and C44 (other malignant neoplasms of skin, active in eight topics), suggesting that they have both large prevalence and a large number of multimorbidity partners belonging to different disease clusters. The top disease codes (i.e., the five with the largest probabilities) in the 10 nonempty topics are consistent with known disease mechanisms (Figure 3B). Specifically, in topic 5, codes E78 (disorders of lipoprotein metabolism and other lipidemias) and I10 are components of the metabolic syndrome,[55] which is associated with increased risk for cardiovascular diseases,[56] an association supported by the three other top disease codes for this topic (I20, I21, and I25, all heart diseases; see Table S2A). Another example is topic 11, whose top codes include four spondylopathy

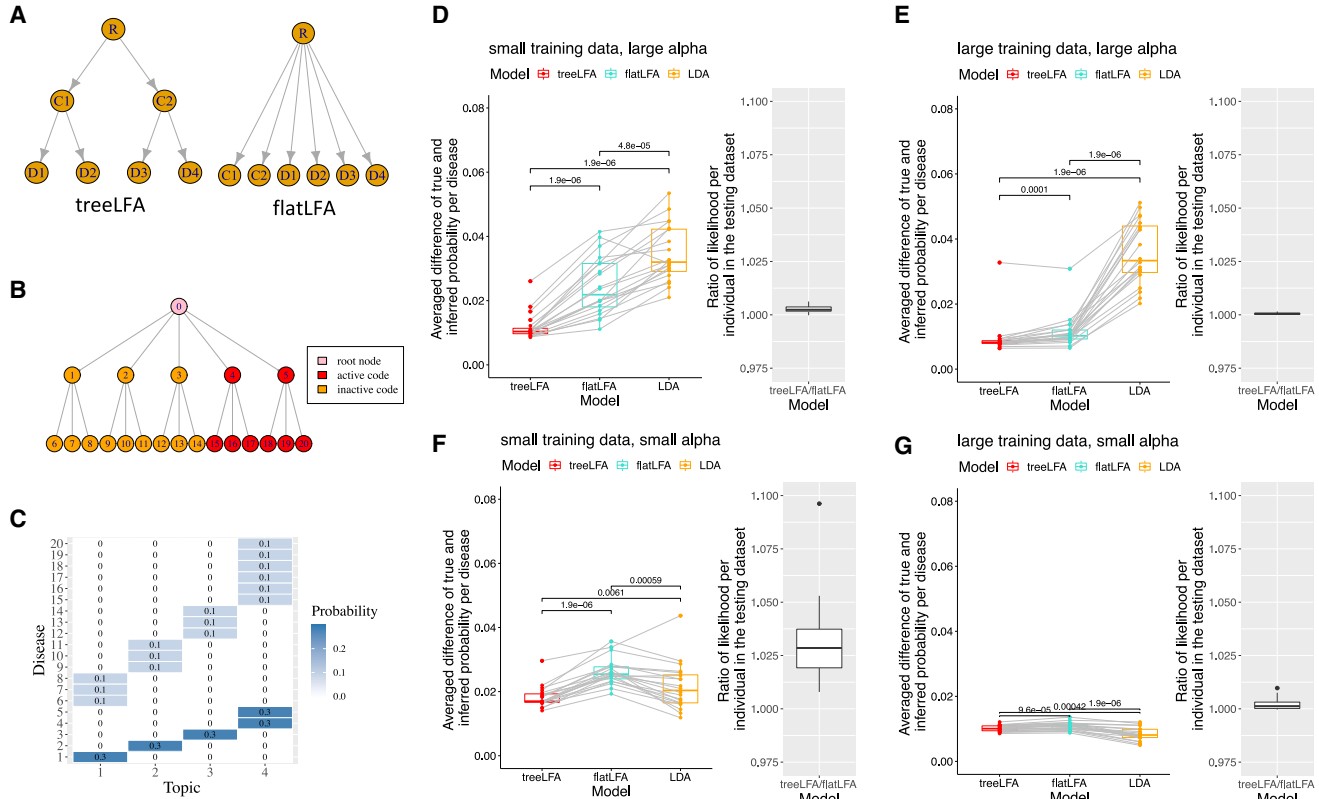

**Figure 2. Comparison of three related topic models (treeLFA, flatLFA, and LDA) on simulated datasets**

(A) The informative and non-informative tree structures of disease codes used by treeLFA and flatLFA.

(B) The tree structure of the 20 disease codes used for simulation. Red nodes on the tree correspond to active disease codes in topic 4 in (C).

(C) The four topics used for simulation. The heatmap shows the probability of 20 diseases in four topics. Disease codes are numbered in the same way as in (B). Inactive disease codes in topics have zero probability.

(D–G) Comparison of three topic models on simulated datasets. The performances of treeLFA, flatLFA, and LDA are compared on four groups of datasets generated using the topics in (C) and different values for D (number of individuals in the training dataset) and $\alpha$ (Dirichlet prior for topic weights). Each group contains 20 paired training and testing datasets. Inference accuracy ($\Delta\varphi$) of the three topic models is shown in the left boxplots, where each dot is the result of one model on one dataset (two-tailed paired Wilcox test for differences in $\Delta\varphi$). The ratio of the averaged per-individual predictive likelihood ($R_{pl}$) for treeLFA and flatLFA is shown in the right boxplots. (D) Results of datasets simulated using D = 2,500 and $\alpha$ = 1. (E) D = 5,000 and $\alpha$ = 1. (F) D = 300 and $\alpha$ = 0.1. (G) D = 1,000 and $\alpha$ = 0.1.

See also Figure S1 and Table S1.

subtypes, while the remaining one is G55 (nerve root and plexus compressions), a common complication of intervertebral disk disorders.

In addition to defining topic vectors, the model also infers individuals' weights for all topics (shown for 2,000 individuals in Figure 3C and Table S2C). Individuals that were not diagnosed with any of the 100 ICD-10 codes (629/2,000) have a weight near 1 for the empty topic, while most other individuals (1,056/1,371) have relatively large weight (>0.1) for fewer than two disease (non-empty) topics, as expected from the sparsity of the data.

To compare treeLFA with other topic models, we used the same input data to train flatLFA, LDA and GETM (graph embedded topic model, an LDA-based model with added functionality; see STAR Methods). We compared the topics inferred by different models (Figure S2 and Table S3A) and evaluated them using topic coherence, topic diversity,[39] and topics' correlation with expert defined disease groups (Tables S3B and S3C).

We also compared the predictive likelihoods of treeLFA and flatLFA (Figure S2B). We found that different models capture the same multimorbidity patterns, but that model structures do play a substantial role in determining the inferred topic structures.

### GWAS on topic weights

We next investigated whether the quantitative traits defined by topic weights can be used to identify genetic variants that are associated with an individual's risk for developing multimorbidities. We performed GWASs on individuals' weights for the 11 topics inferred by treeLFA (topic-GWAS), and the standard case-control GWAS for the 100 ICD-10 codes on the same population for comparison, as well as the 296 Phecodes mapped from these ICD-10 codes (see STAR Methods) to investigate if the choice of coding system for diseases could influence the GWAS results to a large extent.

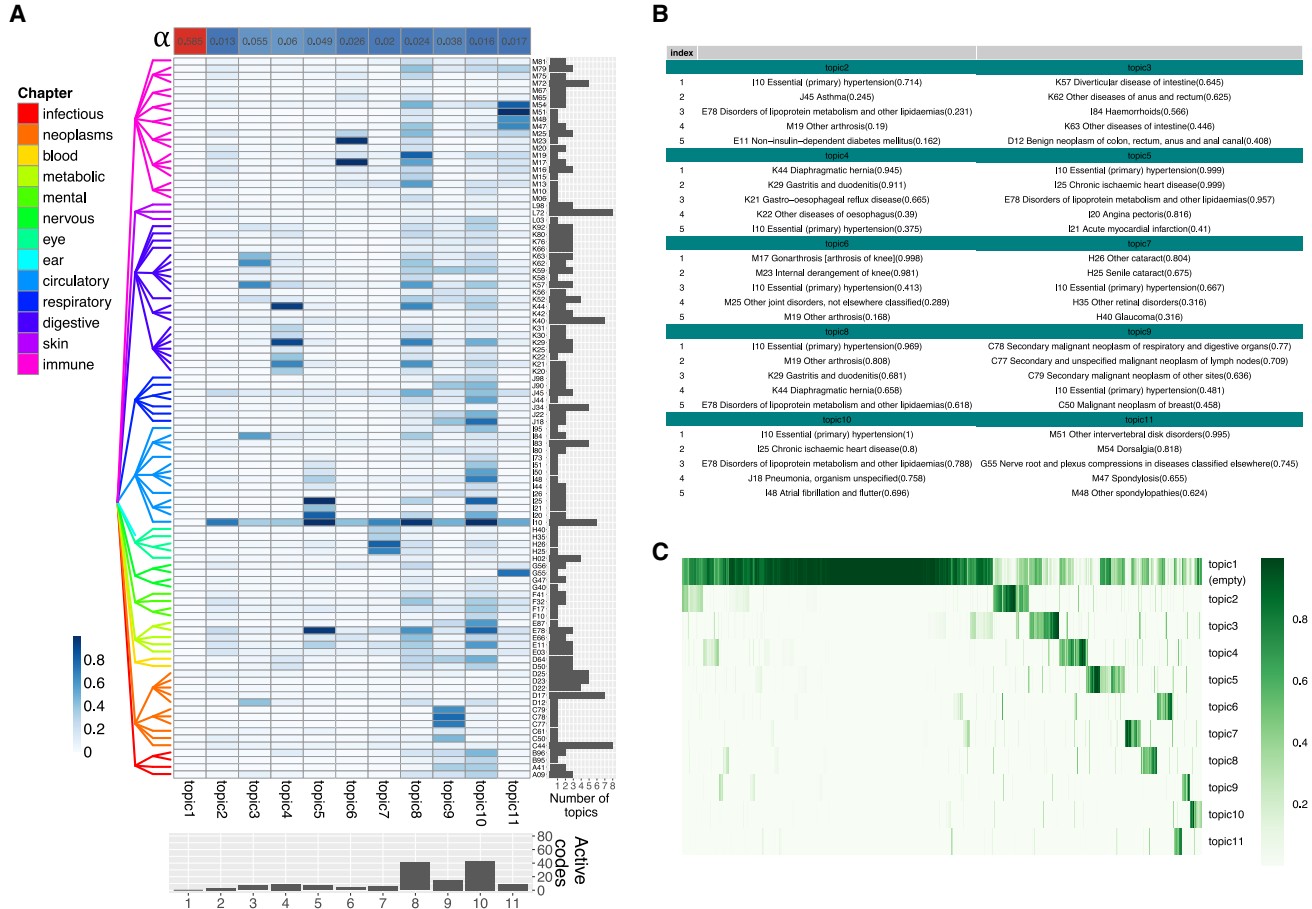

**Figure 3. Inference results given by treeLFA on the top 100 UKB dataset**

(A) Eleven topics inferred by treeLFA from the top 100 UKB dataset. The optimized α (Dirichlet prior for topic weights) is shown as a single-row heatmap on top. The tree structure of the 100 ICD-10 codes is shown on the left. The bar plot below shows the numbers of ICD-10 codes with a probability of at least 0.2 in topics. The bar plot on the right shows the number of topics in which an ICD-10 code is active (with a normalized probability of at least 0.5).

(B) The top 5 codes with the largest probability in the 10 non-empty treeLFA topics (topics 2–11 in A). Numbers in the parentheses are the probabilities of disease codes in topics.

(C) Inferred weights for the 11 topics for 2,000 random individuals. Each column is an individual.

See also Figure S2 and Tables S2 and S3.

From topic-GWAS we found 128 independent loci associated with at least one of the 11 topics, while 812 independent loci were associated with at least one of the 100 ICD-10 codes; 82 loci were shared between the sets (Figure 4A). Phecode GWAS showed similar patterns (Figures S3A and S3B). Breaking this down by topic, we found that unique loci identified only by topic-GWAS were highly non-randomly distributed (Figure 4B and Table S4A). Most unique loci were associated with the empty topic (20 among the 36 topic-associated loci), followed by topic 8 (17/28), which contains a large number (41) of high-probability codes (>0.2) from chapters 11 (diseases of the digestive system, 12 codes) and 13 (diseases of the musculoskeletal system and connective tissue, 13 codes). In contrast, four sparse topics showed no unique loci. Topics 5 (metabolic and heart diseases) and 6 (joint diseases) had many associated loci (50 and 17) and also a substantial number of unique loci (5 and 7). The identification of novel loci indicates that topic-GWAS provides

additional power for discovery in certain scenarios. As an example, Figure 4C (Table S4B) shows that for most of the lead SNPs associated with topic 5, p values of topic-GWAS (for topic 5) are smaller than those of single-code GWAS (for the top 5 active codes in topic 5: E78, I10, I20, I21, and I25). Figure 4D further zooms in on the association signals for two example loci. We consider that the reason only a small fraction of single-code-associated loci were found by topic-GWAS is that most loci are specific to individual diseases, and combining diseases into a topic dilutes the association signal for these loci. We verified this using a simulation and, indeed, found that most singly associated SNPs were not identified as significant when multiple diseases were combined into one trait (Figures S3C and S3D). Despite the limited numbers of topic-associated loci, the genomic control inflation factor $\lambda_{GC}$ and LD score regression (LDSC)[57] indicate that most topics are highly polygenic traits, with the exception of the empty topic and topic 8,

**Figure 4. Topic-GWAS results for the 11 topics inferred by treeLFA**

(A) The total numbers of significant loci found by topic-GWAS for the 11 treeLFA topics and single-code GWAS for the 100 ICD-10 codes and their overlap.

(B) The numbers of significant loci found by both single-code/topic-GWAS and only topic-GWAS for the 11 treeLFA topics.

(C) Comparison of p values given by topic-GWAS for topic 5 and single-code GWAS for the top 5 active codes (E78, I10, I20, I21, and I25) in topic 5 for the same set of SNPs (topic 5-associated lead SNPs).

(D) The Manhattan plot for topic 5, and the regional Manhattan plots for single-code/topic-GWAS results for two example lead SNPs of topic 5.

(E) Comparison of two types of PRS for the 100 ICD-10 codes. One type of PRS is directly constructed using single-code GWAS results. Another is constructed as the sum of individuals' PRSs for topics weighted by the probability of an ICD-10 code in all topics. The AUCs of these two types of PRS on the test dataset are plotted.

See also Figures S3–S7 and Tables S4 and S5.

for which LDSC suggests that uncontrolled confounding factors exist (Table S4C).

We next asked whether topic-GWAS simply identifies loci associated with expert-defined disease clusters, which provide an alternative approach to our data-driven multimorbidity clusters. To address this, we performed GWAS on groups of ICD-10 codes or Phecodes corresponding to internal nodes on the respective ontology. We found that of the 128 topic-associated loci, 41 were not associated with any of the internal or terminal ICD-10 codes; and for Phecodes the corresponding number

was 56 (Figures S3E–S3H). This indicates that topic modeling reveals disease clusters with a genetic basis beyond the expert-driven disease groupings encoded in ontologies. For example, topic 8 has most of its active codes coming from chapters 11 and 13 (see above) and has many unique loci found by topic-GWAS. Interestingly, this disease cluster was also identified by another recent study on multimorbidity using UKB data.[12]

We then compared the topic-GWAS results for topics inferred by treeLFA, flatLFA, LDA, and GETM. Similar numbers of loci were identified by treeLFA and flatLFA (128 and 126; Figure S4A),

while LDA identified many fewer (65; Figure S4B), of which 44 overlap with the treeLFA loci. This difference in numbers of associated loci is mostly due to treeLFA's empty topic (associated with 36 loci), which is not identified by LDA, and also due to differences in the dense topic 8 (treeLFA, 28 loci; LDA, 4 loci) and topics 5 and 6 (Figure S4C). One reason for the relatively poor performance of LDA may be that LDA-derived topic weights are negatively correlated with one another, as they must sum to 1 for each individual, while treeLFA's topic weights are negatively correlated only with the empty topic weight and are otherwise almost independent of one another (Figure S5). GETM found 35 loci, of which 19 overlapped with treeLFA topics (Figure S4D).

## Validation of topic-GWAS results

To exclude the possibility that the unique topic-GWAS associations were driven largely by technical biases or population stratification, we validated the topic-GWAS results in three ways. First, we considered overlap with previously identified loci reported in the GWAS catalog.[58] We found that 114/128 (89.1%) of all topic-associated loci and 36/46 (78.3%) of unique associations have records in the GWAS catalog, and this overlap is consistent across topics (Figure S6A). Second, we compared the functional properties of topic-associated lead SNPs (those not associated with ICD-10 codes) with those of 5,000 randomly selected GWAS tag SNPs (negative control) and all the ICD-10 code-associated lead SNPs (positive control). Using chromHMM-predicted chromatin states as proxy[59,60] (Table S5A [Table S3 of Roadmap Epigenomics Consortium[60]]), we found that, compared with random SNPs, a significantly larger proportion of topic/single-code-associated SNPs are in genomic regions with strong transcription activity (0.44/0.23 vs. 0.18, two-tailed two-proportion Z test, p < 0.05, Bonferroni corrected). In addition, larger proportions of topic/single-code-associated loci are expression quantitative trait loci (eQTLs) (0.77/0.83 vs. 0.5) or have chromatin interactions (CIs) (0.98/0.9 vs. 0.65) in at least one tissue than random loci (Figures S6B–S6D and Table S5B). These analyses show that unique topic-associated loci are functionally comparable to loci identified by single-code GWASs and significantly different from negative controls.

Third, we predict risk of individual diseases on the test data using topic-GWAS results. We first constructed a polygenic risk score (PRS) for topic weights by applying PRSice-2[61] on topic-GWAS results and found that they all show significant association with inferred topic weights on test data (Table S5C). We then used the PRS for topics to construct PRSs for the 100 ICD-10 codes, by adding individuals' PRSs for all topics weighted by the probability of an ICD-10 code in these topics. This can be thought of as decomposing the risk of a disease into several pathways represented by topics. For comparison, we also constructed PRSs for all ICD-10 codes using single-code GWAS results in the standard way. The performance of each pair of PRSs for an ICD-10 code was evaluated on the test data using the area under the receiver-operator curve (AUC) statistic. For 65 ICD-10 codes, topic-PRS AUCs are larger than single-code PRS AUCs (two-tailed two-proportion Z test, p = 0.0037; Figure 4E and Table S5D). This increase was seen most for ICD-10 codes from chapters 5 (mental and behavioral

disorders, 75% [3/4] showing increased AUC), 11 (diseases of the digestive system, 86% [18/21]), and 13 (diseases of the musculoskeletal system and connective tissue, 70% [14/20]). By contrast, single-code PRSs performed well for codes that have a relatively large number of associated loci found by single-code GWAS (>10 associated loci; 18/22 disease codes show larger AUC for single-code PRS). Results given by PRSice-2 were further validated with Lassosum[62] (Figures S7A–S7C). Finally, to make an objective comparison of the topic-GWAS for treeLFA- and LDA-based topic models, we constructed PRSs for ICD-10 codes using the same approach for them, and we found that in the majority of cases (99/100 for LDA and 95/100 for GETM) the PRSs based on treeLFA's results have larger AUCs (Figures S7D and S7E and Table S5E), indicating that treeLFA's topic-GWAS is more informative. Taken together, the three complementary validation approaches indicate that topic-GWAS associations broadly represent true genetic associations with biological phenotypes.

## Inference and topic-GWAS results across treeLFA models

Before applying treeLFA to more diseases, we considered how to select the number of topics (K), a fundamental problem for topic models. We trained treeLFA models with different numbers of topics (K = 2–20, 50, 100) on the top 100 UKB dataset and found that, after clustering posterior samples of topics (STAR Methods), the resulting topics always included an empty topic, and as K increased, the disease topics tended to split into sparser subtopics (Figure 5A) (although some dense topics always remained). We visualized the relationship of all inferred topics across different treeLFA models in a tree by connecting each topic to its most similar topic (measured by Pearson correlation) in the layer above and observed that, as K increased, topics split in a stable way (Figure 5B and Table S6A). These observations indicate that topic-GWAS loci and associated effect sizes should also be stably identified. We verified this for many loci (Figure S8 and Table S6B), and Figure 5B illustrates this for a single variant (Table S6C). We note that for models with K = 50 or K = 100, we infer many near-empty topics, which are unlikely to be stable multimorbidity patterns and are challenging to interpret. Excluding them, we find that the number of distinct topics stabilized at around 25–30 (Figure S9A). The total number of topic-GWAS loci, the number of unique such loci, and the predictive likelihood on the test data (Table S6D) all showed similar patterns beyond K = 20 (Figures 5B and S9B). Taken together, these results indicate that selecting a sufficiently large value for K, combined with *post hoc* clustering of topics, is a computationally efficient strategy for producing a stable and comprehensive set of topics.

## Analyses on hundreds of ICD-10 codes in UKB

We next defined a larger dataset consisting of 436 ICD-10 codes (top 436 UKB dataset, Table S7A). Training treeLFA and flatLFA models with 100 topics, we identified and kept 40 distinct topics (for the convenience of an objective comparison of predictive likelihood; see STAR Methods, Figure 6A, and Table S7B). The 40 treeLFA topics again include several dense topics, many sparse topics (most with one or two ICD-10 chapters enriched

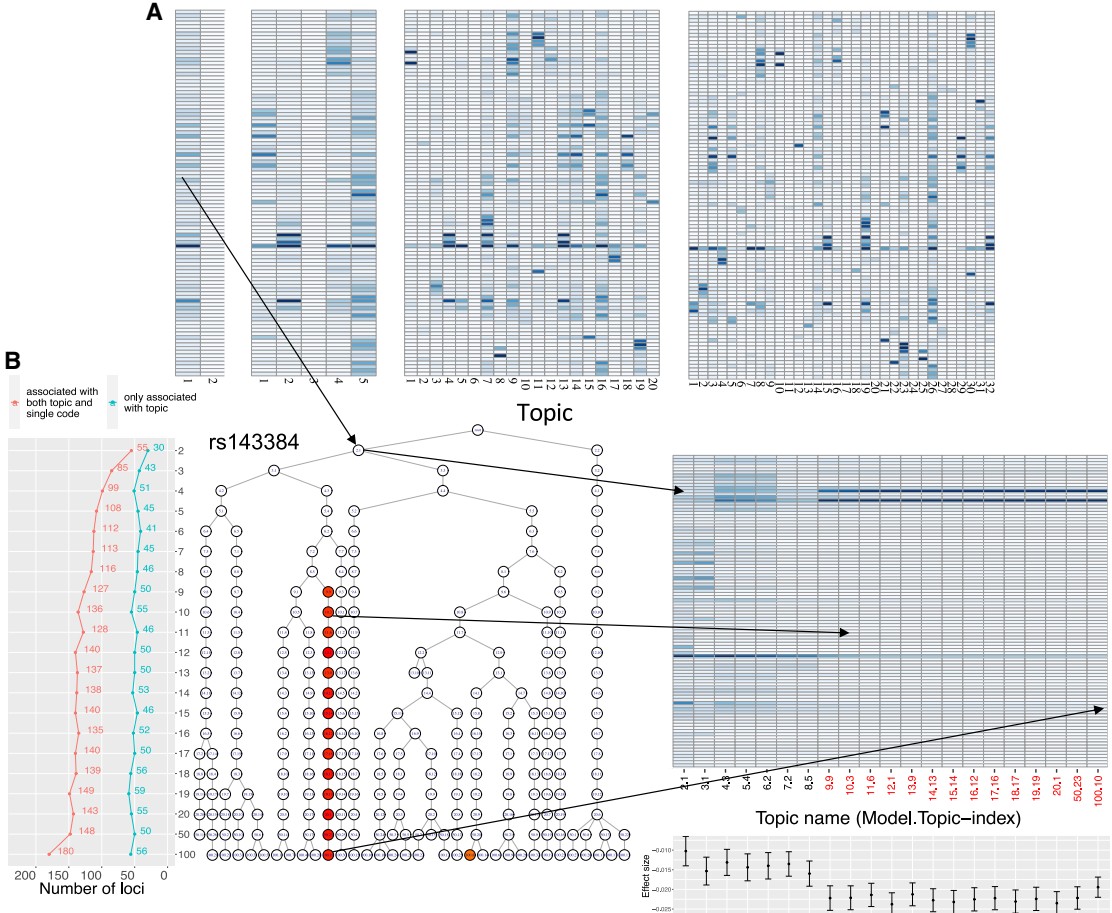

**Figure 5. Inference and topic-GWAS results across treeLFA models set with different numbers of topics**

(A) Topics inferred by treeLFA models set with 2, 5, 20, and 100 topics.

(B) Inferred topics and topic-GWAS results for different treeLFA models. All topics inferred by 21 treeLFA models (set with 2–20, 50, or 100 topics) are organized into a 21-layered tree structure. Each node on the tree corresponds to one topic inferred by one model, and nodes on the same layer are topics inferred by the same model. The "tree of topics" is constructed by connecting each topic on the tree to its most similar one (measured with Pearson correlation) on the layer above. Topics are named as "model index.topic index." Topics significantly associated with SNP "rs143384" (as an example) are colored according to the −log10(p value) given by topic-GWAS. All but one of these rs143382-associated topics are on the same branch of the tree, so all topics on this branch are plotted (heatmap to the right of the tree). In the bar plot below the heatmap, effect sizes and standard errors of rs143384 given by topic-GWAS for the corresponding topics are plotted. The line plot to the left of the tree shows the total numbers of topic-associated loci and, among those, the numbers not found by single-code GWAS for different treeLFA models.

See also Figures S8 and S9 and Table S6.

for active codes, Fisher's exact test, p < 0.05, false discovery rate [FDR] corrected), and a single empty topic (Figures 6A and 6B and Table S7C). The top active codes in these topics (defined as having an unscaled probability >0.3) are shown in Table S8, where topics are annotated based on the categories of these top active codes. For most topics, their top active codes represent similar diseases, such as diseases affecting the same physiological system or having the same pathological mechanism. Comparing topics identified by treeLFA and flatLFA, we found that 32 topics were identified by both models (cosine similarity >0.9, Figure S10A), while the remaining topics have substantial differences. Overall, the predictive likelihood of treeLFA chains was better than that of flatLFA and has a smaller range (Figure S10B and Table S7D).

We then performed topic-GWAS on the 40 treeLFA and flatLFA topics. We found 278 treeLFA (Table S7E) and 260 flatLFA (Table S7F) genome-wide significant loci, with the majority (207) found in both sets and associated with corresponding topics (Figures S10C and S10D). We also performed single-code GWAS on the 436 ICD-10 codes and found 1,093 associated loci; among them, 198 were also associated with treeLFA topics. Lead SNPs for loci associated only with treeLFA topics (80 unique loci) had smaller effect sizes (median absolute effect size 0.021) compared with loci supported by both topics and single codes (0.024) (Figure 6D and Table S7G), indicating that topic-GWAS enabled the discovery of variants with small effects on multiple related diseases. Unique loci were not uniformly distributed across topics (Figure 6E and Table S7H), as in the

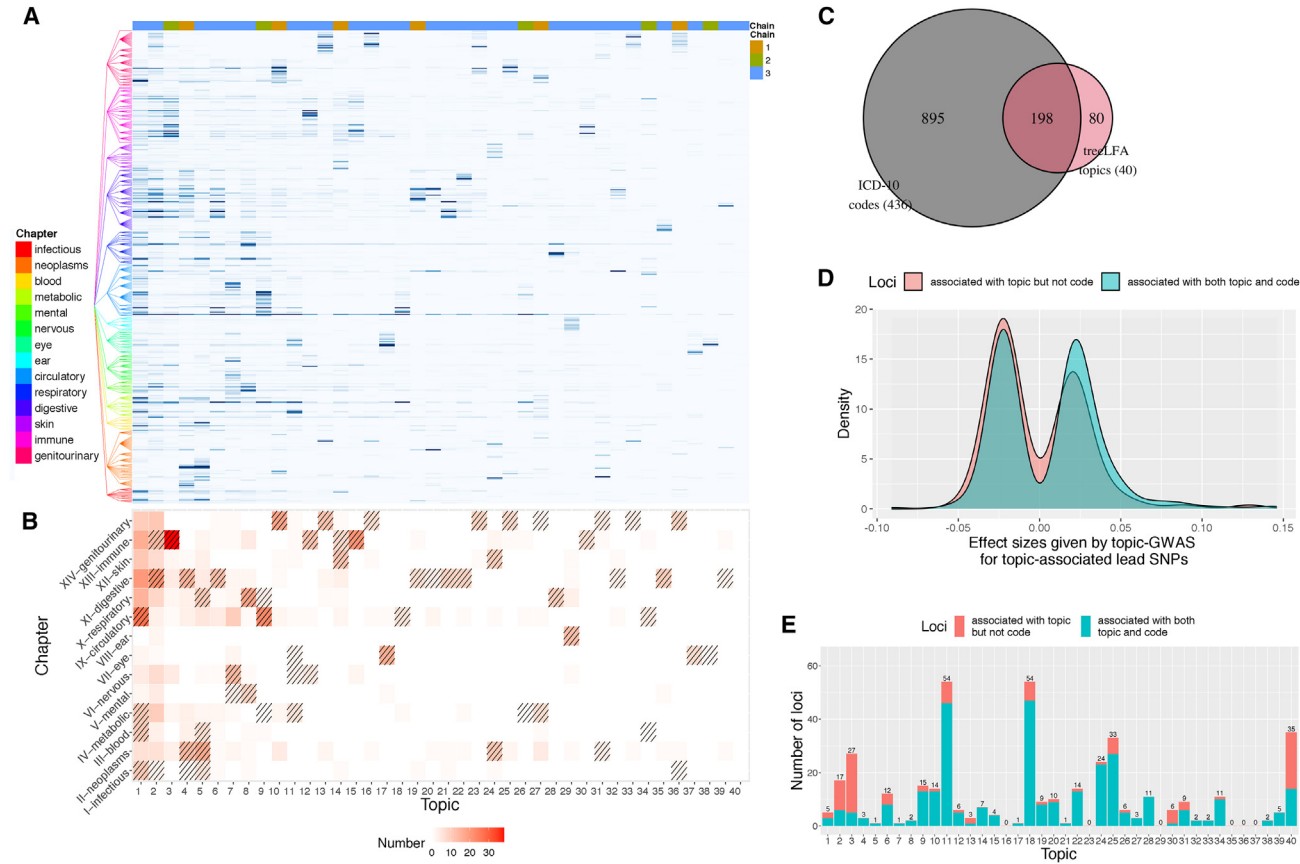

**Figure 6. Inference and topic-GWAS results for the 40 topics inferred by treeLFA from the top 436 UKB dataset**

(A) The 40 topics inferred by treeLFA. Topics are ordered according to their density (the sum of the probability of all codes in the topic). The tree structure of the 436 ICD-10 codes is plotted on the left. The color bar on top shows for each topic the number of treeLFA chains that inferred it.

(B) The numbers of active codes (with a normalized probability of at least 0.5) from different ICD-10 chapters in topics. Enriched chapters for topics are highlighted with shading (Fisher's exact test for the enrichment of 14 ICD10 chapters among 436 ICD10 codes, FDR < 0.05).

(C) The total numbers of loci associated with the 40 treeLFA topics and the 436 ICD-10 codes and their overlap.

(D) Distribution of effect sizes given by topic-GWAS for lead SNPs associated with only topic and lead SNPs associated with both topic and single code.

(E) The total numbers of loci associated with topics (in the same order as topics in A), and the numbers of them not found by single-code GWAS (red).

See also Figures S10–S13 and Tables S7, S8, and S9.

top 100 UKB dataset, many being associated with the empty topic (14 in 35); topics 3 and 30 also had large proportions of unique loci (81.5% and 83.3%), and most of their active codes were from chapter 13 (diseases of the musculoskeletal system and connective tissue). Other topics that were associated with substantial numbers of unique loci are shown in Table S7I. We validated topic-associated loci (Figure S11 and Tables S7J and S7K) and, in addition, those associated with the empty topic and those not in the GWAS catalog (Figure S12 and Table S7L) using the same approaches as with the top 100 UKB dataset, as well as CADD scores[63] and ANNOVAR annotations,[64] and obtained reassuring results. By constructing PRSs for ICD10 codes, we confirmed that treeLFA's topic-GWAS findings can be generalized to test data better than those of flatLFA (Figure S10E), as for 231 of 436 (53%) ICD-10 codes, treeLFA PRSs had larger AUCs on the test data (Table S7K) (two-tailed paired Wilcox test, p = 0.029; two-tailed two-proportion Z test, p = 0.09).

To further interpret the empty topic and its genetic associations, we carried out five secondary analyses. First, we showed that individual's empty topic weights were correlated with their total numbers of diagnosed diseases (Figure S13A). Second, we showed with an example (Figure S13B) that individuals with the same number of diseases but different co-morbidity profiles also have varying weights for the empty topic. Third, we found that the empty topic is strongly associated with variables in UKB reflecting people's overall health condition, such as "overall health rating," and some well-known general risk factors (BMI, smoking, and alcohol drinking) (Table S7M). Fourth, a gene-set enrichment analysis for empty-topic-associated genes (see STAR Methods) indicates lifestyle factors, including sleep duration and regular gym attendance (Figure S13C). Last, tissue enrichment analysis using stratified LDSC (s-LDSC)[65] on GWAS results for the empty topic identified enriched heritability among several central nervous system (CNS) cell types (Figures S13D and S13E and Table S7N). Combined, these

results suggest that analysis of the empty topic provides a data-driven approach to identify SNPs that are related to health behavior phenotypes.

## DISCUSSION

Multimorbidity is a major challenge for today's health-care systems, yet our understanding of it remains limited.[66] Biobanks linked to EHRs present an opportunity for systematic study of multimorbidity and highlights the need for analytic tools for the identification of multimorbidity clusters and downstream analyses with paired omics data.

Here, we developed treeLFA, a Bayesian topic model for binary diagnosis data that admits a prior for topics constructed on existing medical ontologies. We compared it with flatLFA- and LDA-based topic models and found that the prior was effective at extracting relevant topics from limited input data, such as small datasets composed of common diseases (simulated datasets) or large real-world data involving rare diseases (the top 436 UKB dataset). With independent test data, we verified, using both the predictive likelihood and the polygenic risk prediction, that the tree prior improves generalization of the inference, which confirmed the validity of using information in disease classification systems as a supplement to data-driven methods on noisy and sparse real-world data. We also found that treeLFA's model structure better fits the binary input data, resulting in the identification of an empty (healthy) topic and ensuring that topic weights for the remaining disease topics were largely uncorrelated, improving power for downstream topic-GWAS. We also implemented algorithms to optimize hyperparameters of the model and developed a computationally efficient approach to determine the number of meaningful and stable topics.

By applying treeLFA to HES data for 436 common diseases in UKB, we identified 40 topics with varied density reflecting combinations of diseases that tend to co-occur. Most inferred topics likely reflect underlying etiology, as indicated by the fact that the large majority (34/40) show genome-wide significant associations with genetic markers, while 20 topics are associated with 80 novel loci that do not reach genome-wide significance in GWASs for single ICD-10 codes and show evidence of functionality using multiple approaches. The active ICD-10 codes in the topics with the most novel associations are mainly from chapter 13 (diseases of the musculoskeletal system and connective tissue), with substantial contributions also from chapters 4 (endocrine, nutritional, and metabolic diseases), 9 (diseases of the circulatory system), and 14 (diseases of the genitourinary system), suggesting that diseases in these chapters often share genetic risk factors.

We explored constructing PRSs for an ICD-10 code using topic-GWAS results. We found that, for certain codes, especially codes from chapter 13, this new type of PRS outperforms the standard PRS. This improvement in prediction might result from better estimation of variants' effect sizes by topic-GWAS, since treeLFA achieves a dimension reduction from disease space to topic space, which results in fewer traits and therefore more "cases" for each one. These results not only are validation for topic-GWAS, but also indicate the potential gain in utilizing multimorbidity in risk prediction for individual diseases.

In contrast to topic-GWAS, single-code GWAS on the 463 common diseases resulted in 1,093 significant associations, of which the vast majority (895) were not associated with any topic. These genetic analyses indicate that most genetic associations are driven through links to individual diseases. However, most multimorbidity clusters do have a genetic basis, typically composed of pleiotropic genetic variants affecting risks of multiple active diseases in the cluster. In addition, there are also genetic associations for topics that are difficult to identify by single-code GWAS due to the lack of power. From a biological perspective, the loci uniquely associated with topics might reflect pathways causing multiple diseases in the same multimorbidity cluster. It is also possible that different subtypes of a disease have diverse genetic bases; therefore, some loci can be identified by topic-GWAS only where individuals with the same disease and different co-morbidity are separated. Overall, these observations are a helpful starting point for our pursuit of a deeper understanding of the mechanisms underlying multimorbidity clusters.

We consistently identified an "empty" topic, as a result of treeLFA's model configuration. The empty topic provides a way to include healthy people in topic-GWASs, which increases the power for finding associations for other disease topics. In addition, the topic itself showed many genetic associations, many of which (21/35) had not been identified before. It is possible that these variants are associated with risks of multiple diseases, and they were not found before because previous GWASs usually used a negative control group for a specific disease in question, which contained both completely healthy people and people with other diseases. Further analyses of topic-GWAS results indicated lifestyle factors and CNS cell types for the empty topic, a situation similar to the function of genes associated with BMI, obesity, and frailty,[67–69] suggesting that these loci may regulate people's health-related behaviors. Taken together, these results suggest that the empty topic is a biologically meaningful complex and polygenic phenotype related to people's multimorbidity burden and pattern.

There have been many studies aiming at identifying multimorbidity patterns using various methods.[14,17,19,20,70] Most of them focus on dozens of diseases that in addition vary from study to study, making comparisons difficult. We compared the topics identified by treeLFA with multimorbidity networks (interpretable at the level of genetic loci) found by a recent study on 439 common diseases in UKB[12] (433 are included in the top 436 UKB dataset in our study). For most of the reported disease networks, there are corresponding disease topics identified by treeLFA (Table S9). However, the overlap between active codes in treeLFA topics and the disease networks is limited, suggesting significant differences in the inference results. This discrepancy could be caused by the fundamental differences between the two methods, since treeLFA analyzes all diseases simultaneously, while multimorbidity networks were constructed based on pairs of diseases that tend to co-occur. One limitation of topic models is that they cannot determine the relationship between active diseases in the same topic. In contrast, an advantage of topic models is that they make direct use of individual-level data for genetic analyses and allow for making predictions on the test data, which also provides an objective way to compare

different methods. In the future, analysis of multimorbidity should be carried out on more datasets, so that new discoveries can be compared with current findings. Meanwhile, continued efforts in developing advanced algorithms should also be made to uncover more patterns from the rich real-world data.

### Limitations of the study

Our work represents real progress in the understanding of multimorbidity, yet also reveals important unsolved challenges. For instance, while we showed that, taken together, the novel genetic associations likely represent true biology, we have not performed individual replications of the findings in this study in independent datasets, and this may be challenging unless the data sources and methods of data collection are comparable to UKB. In addition, some important factors in modeling diagnosis data, such as errors in assigning ICD-10 codes to individuals and the impact of age on multimorbidity patterns, were not addressed. Overall, the problem of inferring disease topics that are stable, tractable, and biologically meaningful across geographies and health-care systems represents a major challenge for future research.

### STAR★METHODS

Detailed methods are provided in the online version of this paper and include the following:

- KEY RESOURCES TABLE
- RESOURCE AVAILABILITY
  - Lead contact
  - Materials availability
  - Data and code availability
- EXPERIMENTAL MODEL AND SUBJECT DETAILS
- METHOD DETAILS
  - treeLFA
  - Validation of treeLFA with simulated data
  - Inference on the top-100 UKB dataset
  - Genetic analyses
  - Validation of topic-associated loci
  - Analyses on the larger UKB dataset
- QUANTIFICATION AND STATISTICAL ANALYSIS

### SUPPLEMENTAL INFORMATION

### ACKNOWLEDGMENTS

This research has been conducted using the UK Biobank Resource under application number 12788. This work was supported by the Chinese Academy of Medical Sciences (CAMS) Innovation Fund for Medical Science (CIFMS), China (grant number: 2018-I2M-2-002). The research was supported by the Wellcome Trust Core Award grant number 203141/Z/16/Z with additional support from the NIHR Oxford BRC. The research was funded by Wellcome (100956/Z/13/Z to G.M., https://wellcome.org) and the Li Ka Shing Foundation (to G.M., https://lksf.org). We thank Chris Holmes for the discussion.

### AUTHOR CONTRIBUTIONS

Y.Z., conceptualization, data curation, formal analysis, methodology, software, validation, visualization, and writing – original draft; X.J., conceptualization, data curation, methodology, software, and writing – review & editing; A.J.M., conceptualization, project administration, supervision, and writing – review & editing; G.L., conceptualization, investigation, methodology, project administration, resources, supervision, writing – original draft, and writing – review & editing; G.M., conceptualization, funding acquisition, methodology, project administration, resources, supervision, and writing – review & editing.

### DECLARATION OF INTERESTS

G.M. is a director of and shareholder in Genomics PLC and a partner in Peptide Groove LLP. G.L. is a shareholder in Genomics PLC.

### INCLUSION AND DIVERSITY

We support inclusive, diverse, and equitable conduct of research.

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

**Cell Genomics**
**Article**

# STAR★METHODS

## KEY RESOURCES TABLE

| REAGENT or RESOURCE | SOURCE | IDENTIFIER |
|---|---|---|
| **Deposited data** | | |
| UK Biobank | Bycroft et al.[21] | https://biobank.ctsu.ox.ac.uk Data field: 41202, 41204, 22418 |
| ICD-10 coding system | World Health Organization | https://icd.who.int/browse10/2016/en |
| Phecode system | Bastarache[71] | https://phewascatalog.org/phecodes_icd10 |
| GWAS catalogue | MacArthur et al.[58] | https://www.ebi.ac.uk/gwas |
| Summary statistics for topic-GWAS of treeLFA | This paper | https://doi.org/10.17632/rft63p3jcd.1 |
| **Software and algorithms** | | |
| treeLFA | This paper | https://github.com/zhangyd10/treeLFA-demo-CG https://doi.org/10.5281/zenodo.807745 |
| GETM | Wang et al.[39] | https://github.com/li-lab-mcgill/getm |
| PLINK | Purcell et al.[72] | https://www.cog-genomics.org/plink |
| LDSC | Bulik-Sullivan et al.[57] | https://github.com/bulik/ldsc |
| FUMA | Watanabe et al.[73] | https://fuma.ctglab.nl |
| PRSice-2 | Choi et al.[61] | https://github.com/choishingwan/PRSice |
| Lassosum | Mak et al.[62] | https://github.com/tshmak/lassosum |
| sLDSC | Finucane et al.[63] | https://github.com/bulik/ldsc/wiki/Cell-type-specific-analyses |

## RESOURCE AVAILABILITY

### Lead contact
Further information and requests for resources should be directed to and will be fulfilled by the lead contact, Gerton Lunter (g.a. lunter@umcg.nl).

### Materials availability
This study did not generate new unique reagents.

### Data and code availability
This paper analyses existing, publicly available data. The diagnosis data used as input for topic modelling comes from data fields "41202" and "41204" in UK Biobank. The genotype data for individuals in UK Biobank used for GWAS comes from data field "22418" in UK Biobank.

The summary statistics of topic-GWAS for topics inferred by treeLFA on the top-100 and top-436 UKB data sets have been deposited at: https://doi.org/10.17632/rft63p3jcd.1.

The code for treeLFA is available at: https://zenodo.org/record/8174980.

A RMarkdown demonstration for using the algorithm on a small, simulated example dataset is available at: https://zhangyd10.github.io/treeLFA-demo-CG.

Any additional information required to reanalyse the data reported in this paper is available from the lead contact upon request.

## EXPERIMENTAL MODEL AND SUBJECT DETAILS

For this study, we used the diagnosis data and genotype data for 502,366 individuals in the UK Biobank. Among them there are 229,066 males. The average age is 68.46 years ($\pm$ 8.12 *years*). The majority of the population self-identify as British (442,607), the remaining major ethnicity groups include other white backgrounds (16,252), Irish (13,160) and Indian (5,949). Detailed description of the data can be found in the relevant paper.[21]

## METHOD DETAILS

### treeLFA

treeLFA is a topic model developed based on Bayesian mean-parameterized binary non-negative matrix factorization (BNMF), which factorises a Bernoulli probability matrix (used to model the binary input matrix) into a topic matrix and a topic weight matrix. For treeLFA, each topic is a vector of probability, with each entry parameterising the Bernoulli distribution for one disease code. This is fundamentally different from LDA, which only models diseases that have already occurred to individuals using multinomial distributions across all diseases. As LDA, treeLFA models individuals' weights for all topics with Multinomial distributions. treeLFA is a Bayesian model, with a Dirichlet prior put on individuals' topic weights, and two Beta priors put on the probability of active and inactive diseases in topics.

treeLFA incorporates a special prior for topics constructed on a tree structure of disease codes (specified by a medical ontology). This prior assumes that disease codes close to each other on the tree are likely to have similar Bernoulli probability in the same topics. This is achieved by running a Markov process on the tree structure of individual diseases. The Markov process chooses active disease codes (disease codes having large probability in a topic) for each topic by generating binary indicator variables for all disease codes (with 1 represents active codes, and 0 represents inactive codes in a topic) (see Data S1). The two transition probability ($\rho_{01} = P(I = 1|I^{parent} = 0)$, $\rho_{11} = P(I = 1|I^{parent} = 1)$) of the Markov process control the sparsity of topics. Small values for both $\rho_{01}$ and $\rho_{11}$ give rise to sparse topics (most disease codes in a topic are inactive), while large values for both generate dense topics. $\rho_{11}$ also controls the clustering of active codes in topics. With a large $\rho_{11}$, most children codes of an active parent code will be active. As a result, active codes in a topic will gather on the same branch of the tree. On both simulated and real-world datasets, we used small $\rho_{01}$ and large $\rho_{11}$. This reflects our belief that in the real world most topics of disease codes should be sparse (thus we choose small $\rho_{01}$), and that active disease codes in the same topic tend to come from the same subtree (thus we choose large $\rho_{11}$).

The generative process of treeLFA is briefly described as follow:

for k =1…K

  Sample indicator variable $I_{k1}…I_{kS}$ for the S disease codes in topic k using a Markov process

  for s=1…S

    Sample probability variable $\varphi_{ks}$ for disease s in topic k from Beta($a_{I_{ks}}$)

for d=1…D

  Sample topic weight vector $\theta_d$ for individual d from Dirichlet($\alpha$)

  for s =1…S

    Sample topic assignment variable $Z_{ds}$ for disease s of individual d from Categorical($\theta_d$)

    Sample disease variable $W_{ds}$ from Bernoulli($\varphi_{Z_{ds}S}$)

Inference on the treeLFA model is performed using partially collapsed Gibbs sampling,[74] integrating out the topic weight variable $\theta$. The reason we chose Gibbs sampling over variational inference algorithms is that Gibbs sampling was shown to be more accurate,[75,76] especially on a sparse dataset with low signal to noise ratio. We found that the inference accuracy is very important for the association analyses in our study. The disadvantage of Gibbs sampling is that it converges slower than variational algorithms. At present, it is recommended to apply treeLFA to no more than a few hundred disease codes and half a million individuals due to the computational complexity of the inference algorithm (It took about three weeks using 8 nodes on the Biomedical Research Computing cluster in Oxford to perform inference on the top-436 UKB dataset). See Data S1 for more details, including on hyperparameter optimization.

### Validation of treeLFA with simulated data
#### *Overview of three related topic models*

The simulation study served two purposes. Firstly, we aimed to verify that treeLFA can accurately infer latent topics from the diagnosis data encoded as binary variables. Secondly, we aimed to compare the performance of treeLFA to two related topic models (flatLFA and LDA) that are briefly introduced below, such that the influence of model structure and treeLFA's informative prior for topics on the inference can be assessed.

flatLFA has the same model space as treeLFA, with the only difference being that flatLFA uses a non-informative prior for topics, which is constructed on a tree structure where all nodes (representing all disease codes in a topic) are placed directly under the common root node. By comparing the performance treeLFA and flatLFA, the contribution of treeLFA's informative prior for topics to the inference can be assessed.

Latent Dirichlet Allocation (LDA[32])'s model configuration is different from treeLFA. LDA models each disease code diagnosed for an individual with categorical distributions. For LDA, each topic is a categorical distribution (or a Multinomial distribution if the input data is viewed as a count matrix) across the S disease codes, and a Dirichlet prior distribution is used to generate these topics. By contrast, for treeLFA each topic is a vector of S Bernoulli probability for the S disease codes, generated using the Beta prior distributions for active or inactive disease code.

#### *Description of simulated data*

We simulate multiple data sets using different topics and hyperparameters to assess the performance of the three topic models (treeLFA, flatLFA, LDA) in different situations. We simulate the input data sets in two steps. Firstly, we build a tree structure for 20

disease codes (Figure 2B). The tree structure has three layers. The first layer is the root node; the second layer contains five nodes, and each of them has three children nodes in the third layer. Secondly, we generate topics of disease codes according to this tree structure. For our simulation, we manually construct two sets of topics. The first set of topics are likely to be generated using a Markov process with small $\rho_{01}$ and large $\rho_{11}$ on the tree (the hyperparameter setting for treeLFA on the simulated datasets), while the second set of topics are unlikely to be generated by this Markov process. The first set of topics are used to test if the tree structure of codes improves inference accuracy, and the second set of topics test the robustness of treeLFA's inference when the tree structure of codes is wrongly specified. We manually specified the topics to ensure that they are completely distinct from each other and have strong patterns with respect to the clustering of active codes.

Figure 2C shows the first set of topics. Each of the first three topics contains four active codes coming from a different branch of the tree (Figure 2B). The active codes in the last topic come from two different branches of the tree, making it denser than the first three topics. These topics are likely to be generated using a Markov process with small $\rho_{01}$ and large $\rho_{11}$, since a parent code and all its children codes are always in the same state (either active or inactive). In the second simulation setting, active diseases in topics are not generated according to their adjacency on the tree (Figure S1B). As a result, active parent codes always have inactive children codes, and inactive parent codes always have active children codes.

We simulate disease data using the topics described above and the generative process of treeLFA. To evaluate the topic models in different situations in each topic setting, we use four combinations of two hyperparameters to simulate data: $\alpha$ (the concentration parameter of the Dirichlet prior for topic weights $\theta$) and D (the number of individuals in the training dataset). A large value for $\alpha$ means that most individuals will have large topic weights spread across topics. By contrast, a small value for $\alpha$ will make topic weights for each individual more concentrated on a single topic. A large $\alpha$ makes the inference difficult, since most individuals are a mixture of multiple topics. Therefore, data sets simulated using large $\alpha$ require larger D for accurate inference. For each hyperparameter and topic setting, we simulate 20 data sets, with each one containing training and testing data of the same size. Table S1A summarises the hyperparameter and topic settings.

### Implementation of the inference procedure
LDA is implemented using the R package "topicmodels", and collapsed Gibbs sampling is used to perform inference. treeLFA and flatLFA are implemented using the R package "Rcpp" and "RcppParallel" for parallel computation. The randomization for parallel implementation of Gibbs sampling is achieved via Permuted Congruential Generator (PCG64). The code for treeLFA is made available at "https://github.com/zhangyd10/treeLFA-demo-CG" or "https://zenodo.org/record/8174980" (see "Resource availability" section).

For the hyperparameters, we provide the true value of $\alpha$ to train the topic models to simplify the inference tasks and focus on the effects of the tree prior. Beta priors for the probability of active and inactive disease codes in topics ($\varphi$) are Beta(2,4) and Beta(0.3,80). Beta priors for the transition probability ($\rho_{01}$ and $\rho_{11}$) of the Markov process are Beta(4.8,20) and Beta(20,4.8) (treeLFA). For flatLFA only $\rho_{01}$ will be used on the tree, and its prior is Beta(7,20), resulting in approximately the same expected number of active codes in topics as treeLFA. For LDA we try a few different values (0.01,0.1,1) for $\eta$, the concentration parameter of the Dirichlet prior for topics. We find that with $\eta = 0.01$ LDA has the best performance evaluated by the inference accuracy.

For the initialization of hidden variables for treeLFA and flatLFA, we initialise all indicator variables (I) as 0, and simulate probability variables ($\varphi$) using the Beta prior for inactive disease codes. Topic assignment variables (Z) are randomly sampled for all individuals.

For each simulation scenario, ten Gibbs chains were sampled, and 20 posterior samples of hidden variables were collected from each Gibbs chain with an interval of 100 iterations after 15,000 burn-in iterations.

### Evaluation and comparison of topic models on simulated datasets
Two metrics are used to evaluate the models in simulations. The first metric is the inference accuracy, measured with the averaged absolute per disease difference between aligned true and inferred topic vectors:

$$\Delta\varphi = \frac{\sum_{k=1}^{K}\sum_{s=1}^{S}\left|\varphi_{ks}^{true} - \varphi_{ks}^{infer}\right|}{K \times S},$$

where $\varphi_{ks}$ is the Bernoulli probability of disease code s in topic k. To reorder the inferred topics as the true topics, we match each inferred topic to the true topic that has the highest cosine similarity, in a greedy procedure (i.e., once a true topic is matched, it is removed from the matching of the next inferred topic). The two-sided pairwise Wilcox test is used to test for statistical difference between $\Delta\varphi$ of two different models. The second metric is the predictive likelihood on the test data (see Data S1 for more details). For each posterior sample of topics, 200 Monte-Carlo samples of topic weights $\theta$ are used to approximate the predictive likelihood. A sensitivity analysis is done to ensure the number of samples for $\theta$ is enough to have a stable estimate of the predictive likelihood.

### Inference on the top-100 UKB dataset
### The input data for treeLFA
The input to treeLFA includes the diagnosis data for individuals in the UK Biobank, and the tree structure for disease codes. The diagnosis dataset is constructed from the Hospital Episode Statistics (HES) data in the UK Biobank (data fields 41202 and 41204) which is coded using the five-layered hierarchical ICD-10 billing system (ICD-10 tree structure: https://icd.who.int/browse10/2016/en). The first layer of the ICD-10 tree is the root node; the second layer is composed of chapters of diseases (e.g. diseases of the respiratory

system) coded using capital English letters; the third layer contains blocks of disease categories (e.g. acute upper respiratory infections); the fourth layer contains single disease categories (e.g. acute sinusitis); and lastly, the bottom layer contains sub-categories of diseases, which can be, for instance, the same disease occurring at different sites of the human body, or subtypes of the same disease. In UKB, most diagnosed diseases are encoded using codes on the bottom layer (fifth layer) of the tree. We use the fourth layer of encoding as diagnoses, where we replace all diagnoses in UKB with their parental code in the fourth layer on the tree.

The top 100 most frequent ICD-10 codes in UKB from the first 13 chapters of the ICD-10 coding system are chosen to construct the data set. This selection of chapters provides a balance between breadth of phenotype and depth within any one chapter so that the potential benefits of the informative prior of treeLFA can be explored. The diagnosis data is a binary matrix, with each row represents an individual, and each column a disease code. Zeros and ones in the matrix are used to represent the absence and presence of diagnosed ICD-10 codes for individuals. If an individual is diagnosed with the same disease code several times, we keep only one record (See Data S1 for more details). The full UKB data set is randomly split into a training dataset and a testing dataset, containing the diagnosis record for around 80% and 20% individuals. The tree structure of disease codes is encoded in a table with 2 columns: the first column contains all the ICD-10 codes on the tree, and the second column records the parent codes of the corresponding codes in the first column (Table S2A).

### Implementation of treeLFA

The Gibbs-EM algorithm is firstly used to optimise $\alpha$ in two stages. In the first stage we run 2,000 iterations of the Gibbs-EM algorithm. In the E-step of each iteration, we run the Gibbs sampler for treeLFA for 20 iterations and collect one posterior sample of Z (19 burn-in Gibbs sampling iterations before the collection of the posterior sample). In the M-step, $\alpha$ is optimised using this single posterior sample of Z collected in the E-step. In the second stage we continue to run the Gibbs-EM algorithm for 200 iterations. In the E-step of each iteration, we run the Gibbs sampler for treeLFA for 200 iterations and collect ten posterior samples of Z in total (19 burn-in Gibbs sampling iterations before the collection of each posterior sample). The reason to have two stages of training is to balance the computational speed with the inference accuracy. In the first stage, we optimise $\alpha$ more frequently and quickly get close to its optimal value. In the second stage, $\alpha$ is more accurately optimised based on multiple posterior samples of Z.

After optimising $\alpha$, we continue to use the collapsed Gibbs sampler to simulate posterior distributions of all hidden variables (Z, I, $\varphi$, $\rho$), with $\alpha$ fixed at values provided by the last iteration of the Gibbs-EM algorithm. 5,000 burn-in iterations are run before the collection of posterior samples of hidden variables of treeLFA. For each topic model, ten Gibbs chains are constructed, and 50 posterior samples are collected with an interval of 100 iterations from each chain.

To shorten the training with the Gibbs-EM algorithm, we initialise $\alpha$ as (1,0.1,...,0.1). The first entry in $\alpha$ is much larger than the others, which corresponds to the empty topic that will always be inferred from real-world diagnosis data. We also initialise $\alpha$ in other ways, such as using (1,...1), and we find that model converge to the same results regardless of the ways of initialization of $\alpha$, and the optimised $\alpha$ is usually closer to (1,0.1,...,0.1) than other choices. For topic assignment variable Z, we assign the empty topic (Topic 1, corresponds to the first entry in $\alpha$) to all disease variables of individuals without any diagnosed disease codes. For individuals with at least one diagnosed disease code, all topics are randomly assigned to all disease variables. For topics, all indicator variables I are initialised as 0, and probability variables $\varphi$ are randomly sampled from Beta(1,5,000,000). Beta(0.3,80) and Beta(2,4) are used as the prior for $\varphi$ of inactive and active codes. Beta(3,20) and Beta(3,3) are used as the prior for transition probability $\rho_{01}$ and $\rho_{11}$ of the Markov process on the tree. The hyperparameters for flatLFA are set in the same way as treeLFA. For LDA, the concentration parameters of the Dirichlet priors for topic weights ($\alpha$) and topics ($\eta$) are both initialised as a vector of 0.1. $\alpha$ is not optimised since we find that this has negligible influence on the inference result (inferred topics) and downstream analyses (topic-GWAS).

### Post-processing of inference result

The topic weight variable $\theta$ is integrated out during the collapsed Gibbs sampling, therefore their posterior samples need to be approximated using posterior samples of Z and $\alpha$. $\theta$ can be computed as in Griffiths and Steyvers:[74]

$$\theta_{dt} = \frac{N_{dt} + \alpha_t}{N_d + \sum_{k=1}^{K} \alpha_k},$$

where $N_{dt}$ is the total number of disease variables assigned with topic t for individual d, and $N_d$ is the total number of disease variables for individual d.

To combine the inference results given by different Gibbs chains, the "identifiability" issue needs to be addressed, since the order of topics in different posterior samples from different chains may not be the same. We combine all posterior samples of topics from all chains together, and cluster topics before taking the average within each cluster. To cluster topics from all samples, we firstly construct a shared nearest neighbour (SNN) graph using the R package "scran".[72,74] With the SNN graph, we use the "Louvain" algorithm,[71,72,74] a community detection algorithm implemented in the R package "igraph", to assign topics into clusters. After clustering, similar topics coming from different chains or posterior samples will be put into the same cluster. In addition to topics ($\varphi$), we also assign posterior samples of other hidden variables (I, $\rho$ and $\alpha$) to the corresponding clusters according to the clustering result for topics.

The Louvain algorithm is a density based clustering algorithm, which has more flexibility compared to other clustering methods such as K-means since we don't need to specify the number of clusters beforehand, and is less computationally expensive compared

to hierarchical clustering. For the Louvain algorithm, the number of clusters is controlled by the hyperparameter k (the number of nearest neighbours to consider) for the construction of the SNN graph. A large k will result in a small number of clusters, while a small k gives rise to a large number of clusters, though some clusters might be alike. Empirically, we use $k = \frac{N_{ch} \cdot N_{ps}}{2}$, where $N_{ch}$ is the number of Gibbs chains for the same treeLFA model, and $N_{ps}$ is the total number of posterior samples taken from each chain. This relatively small value for k (in theory if there are K different topics as we set for the treeLFA model, each topic should have $N_{ch} \cdot N_{ps}$ posterior samples in total) ensures that different topics are unlikely to be combined into the same cluster in the Louvain clustering. This choice is to balance the total number of clusters found by the algorithm and the uniqueness of different clusters.

As a part of the model selection strategy introduced in this paper, sometimes treeLFA models set with a large and likely excess number of topics are trained, and multiple near-empty topics (topics with few active codes having very small probability) will be inferred. Although the small differences between these near-empty topics are not meaningful, they are usually assigned to different clusters by the Louvain algorithm. To collapse these near-empty topics into a single empty topic, we further apply hierarchical clustering on topics averaged from different clusters given by the Louvain algorithm. During the hierarchical clustering, similar topics are kept being combined until all the remaining topics are significantly different from each other. By visualising the inferred topics using heatmap, one can roughly decide the number of distinct meaningful topics (topics that are not empty or near empty) to keep, and then manually set the number of clusters (topics) to keep for the hierarchical clustering. Alternatively, a threshold for the dissimilarity of topics can be used, such as cosine distance between topic vectors larger than 0.1 or Manhattan distance larger than 1.

### Implementation of GETM

GETM (graph embedded topic model) is regarded as a LDA based topic model in this study since the occurrence of individual words in documents are modelled with Multinomial distributions. A distinguishing feature of GETM is that it incorporates pre-existing knowledge about diseases using pre-trained embeddings for diseases learned with the "word2vec" algorithm from disease ontology. GETM is included in this study to make the comparative analysis for topic models on real world datasets more thorough and comprehensive.

GETM is implemented according to the on-line tutorial (https://github.com/li-lab-mcgill/getm) and the protocol paper for the model.[77] Only conditions (diseases) are provided as input to the model (no medications are provided). The embeddings of all diseases are obtained by running the Node2Vec algorithm using the tree structure of ICD-10 codes.

### Evaluation of inferred topics

1. Topic coherence

Topic coherence[39,77] is used to assess if the top disease codes (with the largest probability) in topics have a high probability to co-occur on the same individuals. It is calculated as:

$$TC = \frac{1}{K}\sum_{k=1}^{K} \frac{1}{S(S-1)} \sum_{i=1}^{S-1} \sum_{j=i+1}^{S} f\left(c_i^k, c_j^k\right),$$

where TC is the topic coherence, K is the total number of topics, S is the total number of disease codes, $c_i^k$ is the $i^{th}$ top disease code in topic k, and f(.,.) is the normalised pointwise mutual information:

$$f(c_i, c_j) = \frac{log \frac{P(c_i, c_j)}{P(c_i) \cdot P(c_j)}}{-log P(c_i, c_j)},$$

where $P(c_i)$ is the marginal probability of disease code i in the training data, and $P(c_i, c_j)$ is the probability of disease codes i and j occuring on the same individuals approximated using empirical counts in the training data.

2. Topic diversity

Topic diversity[39] measures the total number of unique disease codes covered by the top active disease codes of all topics, and is defined as:

$$TD = \frac{U}{10 \cdot K},$$

where U is the total number of unique disease codes among the top 10 disease codes in all topics.

3. Concordance between topics and ICD-10/Phecode groups

To evaluate the alignment of inferred topics and expert defined disease groups, for each topic we calculate its Pearson correlation with all ICD-10 code/Phecode chapters and categories. Chapters correspond to the internal nodes on the second layer of the tree structure of disease codes, while categories correspond to the internal nodes on the third layer of the tree structure. ICD-10/Phecode chapters/categories are represented as binary vectors across all disease codes (for instance, all ICD-10 codes in a chapter will have value 1 in the corresponding binary vector, and the remaining ICD-10 codes will be 0) to calculate their correlation with topics.

treeLFA and flatLFA topics are normalised such that all entries in a topic sum to one for comparison with LDA/GETM topics. For each topic, the top 5 disease groups which have the largest correlation with it are shown.

### Genetic analyses

#### Topic-GWAS and single code GWAS

To find genetic variants influencing individuals' risks for topics of diseases, we perform GWAS using inferred topic weights as continuous traits (topic-GWAS) using PLINK2.[39,73] Since topic weights are real numbers in the range of 0 to 1, the basic assumptions of linear regression do not hold. We apply a logit transformation on topic weights to address this issue before fitting the standard linear model for GWAS. We validate that using logit transformation gives better results than using rank based inverse normal transformation and using no transformation on topic weights. The validation is done by comparing the number of significant loci found by different methods, and the predictive performance of PRS for single codes based on topic-GWAS results (see below). For topic-GWAS, we only include common SNPs (SNPs with a minor allele frequency (MAF) larger than 0.01 in UKB) and individuals who self-report having British ancestry in the training dataset (343,006 individuals in total). Sex, age and the first ten principal components (PCs) of genomic variation are controlled for.

For comparison, we also perform GWAS (logistic regression) using the presence and absence of single ICD-10 codes as binary traits (single code GWAS). The inclusion criterion for individuals, SNPs and covariates are the same as topic-GWAS. In addition to ICD-10 codes, we also use terminal Phecodes mapped from the top 100 ICD-10 codes as traits for single code GWAS. Phecodes are defined by systematically grouping terminal ICD-10 codes into more applicable medical terms based on the judgements of clinicians and researchers,[78] which reduces the granularity of terminal ICD-10 codes. Similar to ICD-10 codes, there is a hierarchical coding system for Phecodes. To map the ICD-10 codes used in the top-100 UKB dataset to phecodes, we firstly extract all terminal ICD-10 codes (on the fifth layer of the ICD-10 tree) that are children codes of the 100 level-4 ICD-10 codes, and then retrieve their corresponding Phecodes according to the Phecode map (https://phewascatalog.org/phecodes) In total, there are 296 terminal Phecodes mapped from the 100 ICD-10 codes.

#### Inflation in P-values given by topic-GWAS

Inflation in P-values are observed for the topic-GWAS results given by all three topic models. The inflation can either be resulted from true polygenicity of the traits (topic weights), or stratification in the population. To differentiate these two possibilities, we carry out LD score regression (LDSC)[57] using the summary statistics of topic-GWAS for all topics. Pre-computed LD scores (based on 1000 Genomes European data) are downloaded and used in the analyses. The genomic control inflation factor $\lambda_{GC}$ and the intercept of LDSC output by the algorithm are compared. A large $\lambda_{GC}$ and small intercept for the same trait suggest true polygenicity causing the inflation in P-values, while large values for both $\lambda_{GC}$ and intercept suggest stratification in the population.

#### Processing GWAS results

To define genomic loci from significant SNPs ($P<5\times10^{-8}$) found by GWAS, we use the clumping function implemented in PLINK-1.9 and the threshold of $r^2>0.1$ to group SNPs in linkage disequilibrium (LD). We define a locus to be an association for both topic-GWAS and single code GWAS as follows: the significant lead SNP found by one GWAS method can be clumped with a significant lead SNP found by the other method.

#### GWAS on internal disease codes on the tree

In addition to grouping disease codes via topic modelling, we also group disease codes completely following the medical ontologies (ICD-10 and Phecode systems). In other words, we use internal codes (such as blocks of categories of diseases and chapters of disease, corresponding to the nodes in the third and second layers of the ICD-10 coding system) of the two disease classification systems as binary traits for single code GWAS. For instance, if both disease codes A and B are under a common parent code C on the tree, then C will be used as the trait for GWAS, and individuals who are diagnosed with either A or B will be used as cases for the single code GWAS for code C. For the 100 ICD-10 codes there are 68 internal codes, and for the 296 Phecodes there are 136 internal codes.

#### Simulation for reduced GWAS power for combined independent traits.

To illustrate the potential reason for only a small proportion of single code associated loci being identified by topic-GWAS, we used a simple simulation. We first simulated five binary genotype variables for 20,000 individuals: $Genotype \sim Bernoulli(0.1)$. Then for each genotype variable, we simulated an associated disease variable: $Disease \sim Bernoulli\left(\frac{1}{1+exp(-b0-b1*Genotype)}\right)$, where $b0 = log\left(\frac{0.1}{1-0.1}\right)$ and $b1 = 0.1$. The simulation of the disease variable was repeated until logistic regression on the disease variable yielded a P-value less than 0.05 for the genotype variable. After simulating the five SNPs and their corresponding associated diseases, we combined the five diseases as one binary trait (individuals having any of the five diseases will have value 1 for the combined disease variable) and carried out logistic regression on the combined disease variable using the five SNPs as predictors. We repeated this simulation 1,000 times and calculated the proportion of single disease associated SNPs that showed significant association (P-value <0.05) with the combined disease variable.

#### Comparison of topic-GWAS results for the three topic models

In addition to topics inferred by treeLFA, topic-GWAS for flatLFA, LDA and GETM inferred topics are also performed. For LDA and GETM, only individuals with at least one diagnosed disease code are used as input for inference. For topic-GWAS, there are two options to deal with the individuals without any diagnosis. We can either exclude them or include them and give them small random weights for all disease topics. We experiment with both methods and find that excluding the completely healthy individuals results in a larger power for topic-GWAS.

## Validation of topic-associated loci

### Validation using the GWAS catalog

We used the GWAS Catalogue[58] (accessed on 2021-11-4) to assess whether topic-associated loci were also found by previous GWAS as significant. We downloaded the full GWAS Catalogue (https://www.ebi.ac.uk/gwas/), and clumped all SNPs in it to topic-associated lead SNPs ($r^2$>0.5 as threshold). If a topic-associated lead SNP can be clumped, it means that a SNP in LD with it was found by a previous GWAS as significant.

### Validation using functional genomic resources

Integrated analysis of GWAS results and functional genomics datasets has gained popularity in recent years.[79] Checking the functional properties of topic-associated loci (such as the enrichment of certain genomic annotations among them) is another approach to validation. We carry out a few functional analyses for topic-associated lead SNPs using the software "FUMA".[80] Since most topics only have a small number of associated loci, we combine all loci (lead SNPs) that are associated with at least one topic and perform analyses on them as a whole. Meanwhile, we also perform the same analyses on all single code associated lead SNPs and 5,000 random SNPs sampled from all SNPs used in the GWAS (the distribution of their MAF is matched to all topic-associated SNPs). Results for these three groups of SNPs are compared. The assumption is that if topic-GWAS find true associations, then these significant SNPs should have an enrichment profile similar to single code associated SNPs (positive control) and different from randomly selected SNPs (negative control).

Three types of functional annotations are used for the validation of topic-associated lead SNPs. First, the three groups of lead SNPs (defined in the above paragraph) are annotated using the 15-core chromatin states predicted by the chromHMM algorithm.[59,60] Since in the 127 available tissues the predicted chromatin states are different, for each genomic locus we use the most active chromatin state (state with the smallest value) across all tissues. Secondly, we calculate the proportions of lead SNPs in the three groups that are eQTL (expression quantitative trait loci) in different tissues using the eQTL mapping function implemented in FUMA, based on dbGaP: phs000424.v8.p2 (eQTL data in the GTEx, version 8). Thirdly, we calculate the proportions of lead SNPs having chromatin interactions with other genomic regions in different tissues, based on the HiC data from GEO: GSE87112 (https://www.ncbi.nlm.nih.gov/geo/query/acc.cgi?acc=GSE87112). The default setting of FUMA for parameters is used in all the analyses. The clumping of SNPs by FUMA uses the phase 3 1000 Genomes European samples as reference, therefore the results are slightly different from the clumping carried out using UK biobank samples as reference. The two-sided two proportion Z-test is used to test for significant differences between the proportions of two groups, with Bonferroni correction done for the 15 chromHMM states.

### Genetic risk prediction based on topic-GWAS results

Another way to validate topic-GWAS results is to use them for prediction tasks on the test data. Because individual variants' effects on traits of interest are usually small, polygenic risk scores (PRS) are constructed to aggregate the effects of tens of thousands of variants. With topic-GWAS carried out on the training data, PRS for topic weights (traits of topic-GWAS) are constructed using the software "PRSice-2"[61], which uses a "C+T" (clumping and thresholding) method, and software "Lassosum"[62], which constructs PRS in a penalised regression framework. No threshold for P-values is manually set for the inclusion of SNPs for PRSice-2, so that the algorithm will try all different thresholds and decide on the optimal one.

We used the testing dataset to evaluate the PRS for topics constructed on the training data. Topic weights of individuals in the testing dataset were inferred by running the Gibbs sampler for treeLFA on them, with $\varphi$ and $\alpha$ fixed at values learnt from the training data (averaged from all posterior samples from all Gibbs chains). Ten Gibbs chains were simulated to infer topic weights for individuals in the testing dataset, and 50 posterior samples are collected from each chain, and their average is used in the subsequent analyses. With inferred topic weights, linear models were fit to evaluate the associations of PRS for topics and the corresponding topic weights, using the logit transformed topic weights as response variables, and PRS for topics as independent variables. The heritability of topic weights is estimated using LDSC as a reference.

To evaluate topic-GWAS results using single code GWAS results as reference, we construct two types of PRS for single ICD-10 codes using single code and topic-GWAS results, respectively. PRS based on single code GWAS are constructed in the standard way. As for PRS based on topic-GWAS, for code s we extract its probabilities in all topics ($\varphi_{ts}$), and calculate an individual's PRS for code s as:

$$PRS_{ds} = \sum_{t=1}^{T}(PRS_{dt} \times \varphi_{ts}),$$

where $PRS_{dt}$ is the PRS of individual d for topic t (constructed using the topic-GWAS result for topic t). The area under the receiver-operator curve (AUC) is used to evaluate the predictive performance of different types of PRS for the same trait on the testing dataset. To compare the topic-GWAS results for different topics models (such as treeLFA and LDA) under a common criterion, we construct PRS for ICD-10 codes using the topic-GWAS results for different models, and compare the AUC of PRS on the testing dataset.

## Analyses on the larger UKB dataset

### The input data for treeLFA

The larger UKB dataset (top-436 UKB dataset) is constructed in the same way as the top-100 UKB dataset, and contains the diagnostic records of the top 436 most frequent ICD-10 codes from the first 14 chapters of the ICD-10 coding system for all individuals in UKB. These codes are all the ones in UKB with a prevalence of at least 0.001 at the date of selection (continued data collection means

that prevalence will tend to increase over time), corresponding to approximately 500 cases. The prevalence threshold of 0.001 is chosen both for computational reasons (this is roughly the limit of what can be performed using available computing resources) and because there must be sufficient occurrences of diseases from which to discover multi-morbidity clusters. As with the top-100 UKB dataset, we partition the full top-436 UKB dataset into training (80%) and testing (20%) datasets. The top-436 and the top-100 datasets use different partitions for the training and testing datasets.

### Inference on the top-436 UKB dataset

The top-436 UKB dataset is more than three times larger than the top-100 UKB dataset, increasing the computational requirements for training topic models. On the top-100 UKB dataset, treeLFA models with different numbers of topics are trained and compared. We find that when we set an excess number of topics for the model, both inferred topics and topic-GWAS results are stable across different models (Figure 5, Table S6A). Therefore, on the top-436 UKB dataset, instead of training many models with different numbers of topics, we train treeLFA and flatLFA models with 100 topics, and cluster and collapse the inferred topics to combine all near-empty topics into a single one.

For the optimization of $\alpha$, the two-stage training strategy with the GibbsEM algorithm is used again. 1,500 iterations are run in the first stage (with a single posterior sample of Z collected in the E-step), and 350 iterations are run in the second stage (with 10 posterior samples of Z collected in the E-step). 50 posterior samples of hidden variables are collected during the last 50 iterations for Gibbs-EM (with an interval of 200 iterations for the Gibbs sampling). For both treeLFA and flatLFA, three Gibbs chains are simulated.

$\alpha$ is initialised as $(1,0.1,\ldots,0.1)$. Beta(0.1,3000) and Beta(1.2,3) are used as the prior for $\varphi$ of inactive and active codes to account for diseases with small prevalence. The remaining hidden variables and hyperparameters are set in the same way as for the top-100 UKB dataset.

We find that different Gibbs chains for treeLFA and flatLFA give slightly different inference results on the top-436 UKB dataset, while different posterior samples from the same chain have a very high level of consistency. Considering the variability among the inference results given by different chains, instead of clustering posterior samples of topics from all chains altogether, we cluster posterior samples from different chains separately. Among the 40 inferred topics, 29 were found by all three treeLFA chains, and five were found by two treeLFA chains, suggesting most topics were stably identified from the data (Figure 6A).

With the averaged $\varphi$ and $\alpha$ for different chains, we calculate their predictive likelihood on the testing dataset, and for both treeLFA and flatLFA we retain the chain which has the largest predictive likelihood, and use its inference result as the input for downstream analyses. For each topic inferred by the chain with the largest predictive likelihood, we check the inference results of the other chains, and annotate the topic with the number of chains that infer them to give a reference of its reliability.

### Topic-GWAS on the top-436 UKB dataset

Since the top-436 UKB dataset is much larger than the top-100 UKB dataset, to increase the inference accuracy for topic weights, after the training with Gibbs-EM algorithm we use Gibbs sampling to re-estimate the topic weights for individuals in the training dataset, which is observed to increase the power of topic-GWAS. $\varphi$ and $\alpha$ are fixed at values averaged from all posterior samples from the chain with the largest predictive likelihood. As a result, there is no longer an identifiability issue, so the results given by different chains (for the re-estimation of topic weights) can be combined directly. For both treeLFA and flatLFA, ten Gibbs chains are used to re-estimate topic weights, and 50 posterior samples are collected from each chain. Topic weights averaged from these chains are used as the input for topic-GWAS.

### Stratified LD score regression (sLDSC)

sLDSC is developed based on LDSC, which partitions the heritability of the trait of interest into different sets of genes to identify relevant tissues and cell types. To find the cell types enriched for the heritability of the empty topic, we run sLDSC on the summary statistics of the topic-GWAS result for the empty topic. We use two sets of cell type annotations to run sLDSC: one gene expression dataset (GTEx) and one chromatin dataset (Roadmap Epigenomics) (for validation, as was done by the developers of sLDSC[81]). sLDSC was implemented according to the on-line tutorial: https://github.com/bulik/ldsc/wiki/Cell-type-specific-analyses.

### Gene-set enrichment analysis for topic-associated SNPs

The software FUMA can find genes that are close to the significant SNPs on the genome (the physical mapping function of FUMA). With the mapped genes, further analyses can be performed. Gene-set enrichment analysis (GSEA) tests for enrichment of different gene sets among a group of genes (e.g. genes mapped from significant SNPs of a trait). We choose genes that are associated with different traits in the GWAS catalogue (e.g. BMI) as the reference gene sets to carry out GSEA for genes mapped from topic-associated SNPs. By doing this, we summarise the major phenotypic associations for topic-associated SNPs found by previous GWAS. The default setting for FUMA is used in all the analyses in this section.

## QUANTIFICATION AND STATISTICAL ANALYSIS

All statistical analyses were performed using R 4.2.2. Methodological details can be found in the STAR Methods, the captions of Figures, and the Result section where statistical tests are used.

