## [Document S2. Transparent peer review records for Zhang et al. · Cell Genomics]

Topic modelling identifies novel genetic loci associated with multimorbidities in UK Biobank

Author list

Yidong Zhang, Xilin Jiang, Alexander J Mentzer, Gil McVean, Gerton Lunter

Summary

Initial submission: Received : December 11th 2022

Scientific editor: Judith Nicholson

First round of review: Number of reviewers: 2
Revision invited : January 25th 2023
Revision received : May 4th 2023

Second round of review: Number of reviewers: 2
Accepted : 7th July 2023

Data freely available: Yes

Code freely available: Yes

This transparent peer review record is not systematically proofread, type-set, or edited. Special characters, formatting, and equations may fail to render properly. Standard procedural text within the editor's letters has been deleted for the sake of brevity, but all official correspondence specific to the manuscript has been preserved.

Referees' reports, first round of review

Reviewer 1

In this paper, the authors address the issue of discovering multimorbidity clusters using a novel method called TreeLFA. They apply the method on a small set of highly prevalent diseases and then on a larger set of diseases in the UKBB. They are able to recapitulate some well-known clusters and are also able to find some new ones. They use the genetic data from UKBB to perform GWAS on the clusters (vs. the individual diseases) and find some hits that they analyze. They also try to validate these hits looking at some genomic characteristics. In general, this is an interesting and well-done paper. It addresses a good problem and brings a novel approach. It has a few issues that I would like to address. With some attention to these thoughts below, I would be enthusiastic about publication.

1. It would be helpful for the authors to address the decision to focus on ICD10 codes as the major results vs. PHECODES. Even though they seem to have done the analysis for both, they stress the ICD results. This seems odd since PHECODES will group together related ICD10 codes and improve power and reduce redundancy, so this choice seems odd and should be explained.
2. The assumption that the ICD10 tree provides an informative prior is not well-justified or obviously true to me. The discussion in lines 182-199 is confusing probably because of this problem with justification. The use of a simulation is a weakness here--because in the simulation the authors build in the assumption that the tree can serve as a useful prior does not address this concern because it is the assumption itself that I have trouble with. Also, the most interesting comorbidities are the ones that are distant (e.g. later in the paper clusters 11 and 13 from different subtrees are mentioned as correlated = that is interesting and not compatible with a "close in the tree means more likely to be co-morbi" assumption). I therefore think that the use of this prior is not well-justified. Indeed, the discussion in 182-199 about the difficulty in distinguishing the results of TreeLFA from the other methods and all the comments about large/small data sets is confusing and can more simply be summarized as "the prior approach didn't work."
3. The authors should discuss the sources of bias and implications of the elderly population of UKBB. This may help them because comorbidities are more likely to be manifest whereas a young population may not show them yet (i.e. they may not be recognized yet). In any case, a discussion of the biases that may result from

the UKBB population would be useful.

4. The whole set of results associated with the "empty topic" is very confusing. It is the major cluster and it has a bunch of genetic hits. Although there is a short paragraph in the discussion about this, I don't feel that it is adequately addressed. What is this empty topic exactly, and shouldn't it be a negative control—why shouldn't the fact that it is so dominant and so "genetically" active be taken as evidence that the method is flawed?

5. Line 220: Similar to (4) above the promiscuity of large and prevalent codes may be a sign of problems with the method?

6. It would be possible to replicate the findings in this paper by looking at claims data from insurance companies (in US) where you are making specific predictions about comorbidities that should also be seen in these databases. This might add to the paper, but I recognize would also be considerable extra work.

7. Lines 237-245 make me really wonder if TreeLFA is adding anything since the other methods seem to get essentially the same results. This should be considered and discussed further.

8. Lines 248-254 raise the issue that the topic-GWAS generally will have a bigger N and more power, and this is never addressed (as far as I can see) in the paper. It might be useful to disentangle the effect of bigger N for topics vs. their lack of specificity, in order to make more fair comparisons with the individual code-GWAS.

9. I think the authors are missing an opportunity to highlight the main interesting findings; the discussion of the GWAS is fairly clear but does not address the key question directly: what are the SNPs that are topic-specific but NOT found in the code-GWAS (even after correction for improved power in the topic-GWAS)? These are the SNPs that will implicate genes and loci that might not be strong for individual diseases but are strong in the context of the topic. These are the main discoveries of interest and they are not sufficiently highlighted in this paper. Indeed, the validation strategy of looking at GWAS catalog (e.g. line 308 ff) is counter-productive because we are interested in those SNPs that would NOT be in the GWAS catalog because they are only "visible" in the context of the topic-GWAS and so this is a big lost opportunity that should be addressed in a revision.

10. Discussion around line 270 again is confused by the different power of topic-GWAS (more power) vs. code-GWAS (less power) and this must be addressed somehow in order for this discussion to make sense.

11. Discussion on lines 279-293 further makes me wonder if the claim of tree-based prior makes any sense or improved the performance in any meaningful

way.

12. The discussion on lines 316-323 is not convincing because those observations are true for virtually all GWAS hits that have ever been observed (as far as I can tell), and so this is a weak validation.

13. The PRS calculations are useful and interesting, but again the AUCs should not be compared directly until something is done to normalize for the increased power of the topic-GWAS hits, etc...same issue as discussed above.

I shared this manuscript with a senior graduate student working in this area. I append their comments here, in case also useful to the authors or editors.

Thank you for the opportunity of reviewing this paper. The authors identify comorbidity clusters - "topics" in their definitions - by treeLFA and a prior ICD-10 coding system. Then, they present the GWAS results on the topics compared to those of constituent disease codes. The authors aim to show how topic modeling help discover multimorbidity patterns. Still, they need to provide more statistical justification on how the genetic analysis of the combined traits provides insights beyond the study of individual code based on their GWAS. Overall, the work is well-written and covers a great topic of genetics involved in multimorbidity pathogenesis that deserves more attention.

Major issues

1. The use of ICD-10 could be error-prone. It concerns me that authors have discussed little the previous work related to ICD-10 and clinical applications of ICD-10. ICD-10 coding could be inaccurate and associated with issues such as missingness. These problems are likely connected to a frequently assigned "empty" topic that needs further discussion.
2. Some common ICD-10 codes were not covered in Table 1, type 1 diabetes mellitus E10, for example. E11 (type 2 diabetes) was found in many disease topics, but no mention of E10 in any metabolic-related clusters. How is the number of 100 top codes decided to use instead of 120, 150, and 200? In addition, defining the disease clusters by parameter tuning (K) could be noisy. The optimal K is data size-dependent and involves more post-hoc selections if including more diagnostic codes.
3. The authors leverage the ICD-10 architecture. ICD-10 structure is a classification of diagnostic codes which does not explicitly consider comorbidities. Can authors describe more about how the hierarchical structure of the ICD-10 codes could be

helpful to them? What are the implications associated with good performance of using this outside information?

4. GWAS on topic weights makes sense but needs a more thorough analysis of why this could be useful to disease studies. The difference between the number of loci found by 11 topic-GWAS (82+46) and the single code GWAS (730) is significant and surprising, same big difference for the top-436 dataset. Can authors address why the topic-GWAS has only identified a small proportion of unique loci found in the single code GWAS? How specifically does the relative prevalence of each disease code make an impact on the topic-GWAS? They need to adequately address these issues to conclude that topic-GWAS can provide suitable power for genetic discovery underlying the comorbidity patterns. Authors used the higher p-values of lead SNPs for certain diseases in the topic-GWAS to indicate their "satisfying" statistical power. This can be over-optimistic without addressing the low number of identified loci. Comparing the results to those of individual GWAS in depth would be more helpful. It is reported that 89% and 78.9% of unique loci and variants identified in their GWAS are in the GWAS catalog. The authors discussed little of the remaining associations found in the topic-GWAS but not in the GWAS catalog. Are those potentially true signals or noises? Also, the authors should note that the GWAS catalog is incomprehensive and outdated. All the GWAS stats should be included in the supplementary.

5. Comparing topic-GWAS to the phecode-oriented GWAS needs to be clarified. Phecodes are inclusive and correspond to many ICD-10 codes that might not be present in the same "disease topic".

6. For PRS, authors should report the statistical numbers on if topic-PRS is significantly higher than single-code PRS (t-test, etc. on all codes) for both top-100 and top-436 datasets. Can authors explain why they add individuals' PRS for the topics weighted by the probabilities of the ICD-10 codes of interest in each topic? How important is this?

Minor issues:

1. The larger dataset usually refers to a higher number of individuals in the study. It should be more precisely stated (e.g., analyses on the top-100 and top-436 datasets).
2. As briefly introduced by the authors, many other ways of extracting comorbidity patterns exist. A few more citations on those should be sufficient.
3. How efficient is treeLFA approach? How computationally expensive is it?

Reviewer 2

Authors present a topic model called treeLFA with 3 main differences in contrast to the well-known Latent Dirichlet Allocation (LDA) model: (1) to model binary code in the cross-section EHR data such as the HES in UK Biobank, Bernoulli likelihood was used instead of Categorical likelihood in LDA with conjugate prior changed from Dirichlet to Beta for ϕ ; (2) the index for the hyperparameters "a" for the topics-diseases Beta distribution $\phi \sim \text{Beta}(a_{i,0}, a_{i,1})$ are further parameterized by Bernoulli variable "i", which follows Markov process dictated by the transition probabilities ρ along the known ICD-10 taxonomy. Overall, the paper is clearly written with a good organization. The technical details were also fully presented in the "Analytical notes" and easy to follow (given my expertise in topic modeling of administrative data such as the ICD codes). However, I still have quite a few comments.

Major comments:

1. There have been several recently developed topic models for modeling EHR data with more efficient stochastic variational inference algorithm compared to the Gibbs sampling algorithm. For example, MixEHR infers multimorbidity by modelling several types of EHR data (Li et al., Nat Comm 2022). On the technical side, variational inference as implemented in the MixEHR will resolve the topic switching issues during the MCMC sampling since one only needs to take the converged topics at the end of the iteration. I suggest author compare treeLFA with MixEHR or at least cite the paper as a related method.
 - a. Li, Y. et al. Inferring multimodal latent topics from electronic health records. Nat Commun 11, 2536 (2020).
2. In terms of using disease taxonomy to model UKB data, a more recent approach called graph-embedded topic model uses node2vec to learn the embedding of the disease codes and then leverage that to infer topics (Wang et al 2022, 2023). Please mention this method as a related method or better yet compare it with your treeLFA
 - a. Wang, Y., Benavides, R., Diatchenko, L., Grant, A. V. & Li, Y. A graph-embedded topic model enables characterization of diverse pain phenotypes among UK biobank individuals. iScience 25, 104390 (2022).
 - b. Wang, Y., Grant, A. V. & Li, Y. Implementation of a graph-embedded topic model for analysis of population-level electronic health records. Star Protocol 4, 101966 (2023).
3. When aggregating the frequency of ICD-10 codes over multiple visits, the UKB data can still have count data. In line 787, author mentioned that they keep only

one record if that happens. How frequent is this? This also shows the inadequacy of using Bernoulli as likelihood instead of binomial.

4. A more technical issue is the computational expense: the Bernoulli likelihood requires computing the entire ICD-10 codes vocabulary for each subject whereas LDA only models the observed words. As the second premise in the introduction, the treeLFA model is set to tackle sparsity of ICD-10 codes. Yet the authors focused only on the top 100 or top 436 most frequent ICD-10 codes. Why not model *all* the ICD-10 codes and removing only the low frequent ICD-10 codes (e.g., the ones that occur in fewer than 5 subjects)? If scalability is the problem (as authors mentioned given the limited computing resource), please provide time complexity and clearly describe in a separate section called "The limitation of the study" as treeLFA does not scale to more than 450 ICD-10 codes. Indeed, the scalability issue is due to the fact that you model each code as Bernoulli, which force you to model *all* input ICD-10 codes for *each* subject as opposed to modeling only the *observed* ICD-10 codes for each subject as in the LDA.

5. Line 233: "Individuals that were not diagnosed with any of the top-100 ICD-10 codes (629/2,000) have a weight near 1 for the empty topic". My question is why include those "empty documents" in the first place?

6. Line 255: besides showing the number of loci, what are the heritability estimates for those topics when treated as phenotypes?

7. Line 260: how to explain that most unique loci were associated with the empty topic? I found this rather counter intuitive.

8. Line 287: "insights" is an overstatement without biological supports since several topics (other than the examples of topic 8 and topic 13) are not directly mapped to a single disease theme.

9. Please compare the predictive likelihood on the 20% test subjects among treeLFA, flatLFA, and LDA using both top-100 and top-436 UKB test data. I am aware that the authors have compared treeLFA with flatLFA in Supplementary Figure 10b as they described in line 295. However, this is an important experiment on real data and more direct compared to the AUC shown in Figure 4e (which is also valuable in its own light). I suggest do a thorough comparison among the 3 methods on the predictive likelihood and show them as a sub panel in Figure 4 in the main text. Also, please perform the comparison using the top-100 UKB data.

10. Please also compute topic concordance to evaluate how the inferred topics ϕ concord with the expert-curated PheCodes or Clinical Classifications Software (CCS) codes: for each topic, use the highest topic coherence scores over all ICD

groups to represent it; compare the distribution of those scores for flatLFA topics and those scores for LDA topics.

11. Line 416: The authors did not provide any interpretation of this seemingly rich figure. What are the biological implications of these enrichments? how to explain cases where the random loci give higher enrichment (e.g., chromHMM state 5 and 9)? Please elaborate on the findings of this figure.

12. Line 427: For Supplementary Figure 10e, what's the p-value based say Wilcoxon signed rank test or KS-test? They don't look significantly different.

13. Line 468: PRSice-2 is often a baseline PRS method compared to LDpred2, Lassosum, SBayesR. Please show the PRS AUC using those methods to further home in this contribution.

14. Line 653: the Analytical Notes are important details as integral of the manuscript. Given that this is a method paper, I suggest adding the model description to the Methods section in the main text. I also have quite a few comments on it as described below.

15. Line 700-702: This well written paragraph should be added to the model description as the rationale for *using* Markov process. The simulation just follows the same data generative process assumed by your treeLFA.

16. Supplementary Figure 9b does not look like saturate to me.

17. Line 717: in the simulation, are the first 3 topics the same or they just have the same Beta hyperparameters? Please clarify.

18. Line 741: Why not use the fixed point in Eq 7 to estimate alpha?

19. The software page is incomplete. R markdown does not show up on the github browser.

20. Topic identifiability. I invite authors refer to MixEHR-Guide and MixEHR-Seed for solving topic identifiability issue using a expert-curated guide (i.e., PheCodes).

a. Zhang, A. et al. Automatic Phenotyping by a Seed-guided Topic Model. Proc 28th Acm Sigkdd Conf Knowl Discov Data Min 4713-4723 (2022)
doi:10.1145/3534678.3542675.

b. Ahuja, Y., Zou, Y., Verma, A., Buckeridge, D. & Li, Y. MixEHR-Guided: A guided multi-modal topic modeling approach for large-scale automatic phenotyping using the electronic health record. J Biomed Inform 134, 104190 (2022).

21. Line 849: Why not use K-means with K set to be the same of the number of topics (i.e., $K=T$)? Also, some similarity threshold is needed since these clustering algorithm always form cluster regardless how similar the topics are in the same cluster.

22. Line 857: Lots of post-hoc processing is done here, which undermines the

original model elegance. It leaves the impression that the treeLFA model does not seem to work well and requires a lot of massaging at the end.

23. Line 873: While the logit ($\log \theta / (1 - \theta)$) makes sense, Dirichlet regression may fit better.

24. Analytical Notes Line 40 for the choice of Beta priors: one benefit of LDA is to have a simplex Dirichlet over each topic such that the posterior topic distribution (also a Dirichlet due to conjugacy to the multinomial likelihood) will sum to 1 over the ICD-10 codes. This naturally yields diverse sets of topics. However, by having the Beta prior, each ICD-10 code under each topic sum to 1 over the active and inactive state. As a result, the topic distribution is much denser (since each topic does not sum to 1 over the ICD-10 codes). This may substantially decrease the topic interpretability and diminish the benefits of topic modeling. As a common quantitative metric to evaluate the topic quality, authors can evaluate topic diversity and topic coherence in comparison with LDA.

25. Qualitatively, I suggest displaying the top 5 active codes per topic for a select disease topics from their treeLFA and display their active state probabilities together in a heatmap. This is to show topic diversity and whether the top codes are meaningful under the topic (e.g., similar diseases should group together).

26. Can authors comment on the human bias in assigning the ICD-10 codes to each subject? For example, Song et al (2021) shows the benefits of using expert-specific topic distribution to model the expert domain knowledge.

a. Song, Z. et al. Supervised Multi-Specialist Topic Model with Applications on Large-Scale Electronic Health Record Data. in Proceedings of the 12th ACM Conference on Bioinformatics, Computational Biology, and Health Informatics 1-26 (Association for Computing Machinery, 2021). doi:10.1145/3459930.3469543.

Minor comments:

1. Line 214: should be "scale" instead of "normalize"

2. Line 297: "many fewer" => "much fewer"

3. Line 298-300: This does not seem to a disadvantage of LDA to me since the empty topic is not useful anyway and the dense topic 8 is hard to interpret compared to the sparse topics. Negatively correlation between topics imply topic diversity and having topic distribution sum to one over all codes imply sparsity. Neither is a unfavorable property of LDA.

4. Following my above recommendation, Figure 1 and 2 in the Analytical Notes should be added as subpanels to Figure 1 in the main text.

5. Analytic Notes line 55-57: For the completeness, please add: $\rho_{10} = 1$ -

ρ_{11} and $\rho_{00} = 1 - \rho_{01}$ corresponding to from active-to-inactive and inactive-to-inactive transition.

6. Please describe in the Appendix Figure 2 panel A legend that I_{ts} index the shape parameters "a" for Beta variable ϕ_{ts} (as shown in Equation 1 of the data generative process and Equation 3 for the conditional of ϕ_{ts}).

7. In Analytic Notes, Please add below Equation 3 that "Equation 3 was derived based on the fact that Beta is conjugate to Bernoulli likelihood."

8. In Analytic Notes line 137: "in topics" => "for all topics".

9. Lastly, the manuscript is long -- there are 78 PDF pages. While I don't think there are redundant content, it is difficult to navigate without hyperlink sections. Please add those to help the reviewing process.

Authors' response to the first round of review

Reviewer #1:

In this paper, the authors address the issue of discovering multimorbidity clusters using a novel method called TreeLFA. They apply the method on a small set of highly prevalent diseases and then on a larger set of diseases in the UKBB. They are able to recapitulate some well-known clusters and are also able to find some new ones. They use the genetic data from UKBB to perform GWAS on the clusters (vs. the individual diseases) and find some hits that they analyse. They also try to validate these hits looking at some genomic characteristics. In general, this is an interesting and well-done paper. It addresses a good problem and brings a novel approach. It has a few issues that I would like to address. With some attention to these thoughts below, I would be enthusiastic about publication.

Response:

We thank the reviewer for the supportive comments. In the revised version of the paper we added new analyses aiming to further validate and interpret the empty topic. We obtained convincing validation results, and discussed the potential biological mechanisms involved. We also added validation analyses specific to those topic-associated loci that are not in the GWAS catalogue. We clarified the justification of using the tree-structure prior for topics, and the comparison between different GWAS methods (single code GWAS on ICD-10 codes/phecodes and topic-GWAS), and added a simulation to show the reason why only a small fraction of single code associated loci were found by topic-GWAS. Technically, we supplemented a computational cost figure and information about the scalability of the algorithm. The other concerns were also addressed accordingly.

Comment 1

It would be helpful for the authors to address the decision to focus on ICD10 codes as the major results vs. PHECODES. Even though they seem to have done the analysis for both, they stress the ICD results. This seems odd since PHECODES will group together related ICD10 codes and improve power and reduce redundancy, so this choice seems odd and should be explained.

Response:

We agree Phecodes could be a better coding system to represent disease ontology. We performed our analyses on ICD-10 codes as ICD-10 is a more widely used classification system than Phecode, therefore we considered the ICD-10 system to be a better starting point to evaluate our new treeLFA methods.

We have edited the text to clarify that the aim of performing single code GWAS on Phecodes is to validate the patterns of topic/single code associated loci we discovered with ICD-10 data, and to investigate if the coding system used in analysis can largely influence the single code GWAS result. Changes: Manuscript line 256-258, 262-263.

Comment 2

The assumption that the ICD10 tree provides an informative prior is not well-justified or obviously true to me. The discussion in lines 182-199 is confusing probably because of this problem with justification. The use of a simulation is a weakness here--because in the simulation the authors build in the assumption that the tree can serve as a useful prior does not address this concern because it is the assumption itself that I have trouble with. Also, the most interesting comorbidities are the ones that are distant (e.g. later in the paper clusters 11 and 13 from different subtrees are mentioned as correlated = that is interesting and not compatible with a "close in the tree means more likely to be co-morbi" assumption). I therefore think that the use of this prior is not well-justified. Indeed, the discussion in 182-199 about the difficulty in distinguishing the results of TreeLFA from the other methods and all the comments about large/small data sets is confusing and can more simply be summarised as "the prior approach didn't work."

Response:

We have edited the text to clarify the rationale of using the tree-structured prior for topics, and its effects. The tree prior was used to incorporate information in the currently accepted disease classification systems (such as the ICD-10/Phencode systems) into the modelling framework, and they represent our knowledge about the relationship of diseases. When there isn't enough data to infer stable multimorbidity patterns, we assume that multimorbidity patterns are likely to be in alignment with the structure in the disease classification system.

As for the evidence of the tree prior improving inference, on the top-436 dataset we compared treeLFA and flatLFA using predictive likelihood and PRS for single codes, and found that treeLFA had advantages over flatLFA evaluated using both metrics (Supplementary Figure 10B,E). These results further verified that using the tree prior improved the inference, though might not to an extent as large as we once expected.

Changes: Manuscript line 149-151, 494-498.

Comment 3

The authors should discuss the sources of bias and implications of the elderly population of UKBB. This may help them because comorbidities are more likely to be manifest whereas a young population may not show them yet (i.e. they may not be recognized yet). In any case, a discussion of the biases that may result from the UKBB population would be useful.

Response:

A discussion about currently unsolved problems in finding multimorbidity patterns using treeFLA has been added, including the influence of age on multimorbidity.

Change: Manuscript line 576-578.

Comment 4

The whole set of results associated with the "empty topic" is very confusing. It is the major cluster and it has a bunch of genetic hits. Although there is a short paragraph in the discussion about this, I don't feel that it is adequately addressed. What is this empty topic exactly, and shouldn't it be a negative control—why shouldn't the fact that it is so dominant and so "genetically" active be taken as evidence that the method is flawed?

Response:

The inference of the empty topic is a feature of treeLFA, as a result of modelling diseases with Bernoulli distributions. This is fundamentally different from LDA, for which the inference of an empty topic is not possible due to the definition of topics (as categorical distributions over all diseases). We have clarified in the discussion section that this property of treeLFA is advantageous as it improves the power of the downstream GWAS, and analysed the reason why it has many (novel) associations.

We have carried out additional analyses to show that the empty topic represents people's multimorbidity burden (related to both people's total number of diagnosed diseases and comorbidity structure). We have also done validations specific for the empty-topic associated loci, and obtained convincing results. We also emphasised that many empty topic associated loci were found to be associated with people's lifestyles by previous studies, and its heritability is enriched in CNS, a situation similar to other relevant phenotypes such as frailty. All these new results demonstrate that the empty topic should be a meaningful phenotype.

Changes: Manuscript line 440-464, 541-554, 1197-1203; Supplementary information line 170-186, 189-205, 301-302, 351-356; Supplementary Figure 12, 13; Supplementary Table 23, 35-36.

Comment 5

Line 220: Similar to (4) above the promiscuity of large and prevalent codes may be a sign of problems with the method?

Response:

We consider the promiscuity of prevalent disease codes, in particular hypertension, indicates this phenotype may be less a classical “disease” (relatively uncommon and chance occurrence modulated by genetics and lifestyle) and more a general component of ageing (although still modulated by lifestyle and genetics), which decouples it from clusters of correlated uncommon phenotypes.

Comment 6

It would be possible to replicate the findings in this paper by looking at claims data from insurance companies (in US) where you are making specific predictions about comorbidities that should also be seen in these databases. This might add to the paper, but I recognize it would also be considerable extra work.

Response:

We fully agree that analysing another dataset to see if the same multimorbidity patterns can be found would be an excellent addition and this is certainly a must-do in the future. However, the work required is very substantial (comparable to the work carried out to date) and our preference is to leave this to the next stage of the research.

Comment 7

Lines 237-245 make me really wonder if TreeLFA is adding anything since the other methods seem to get essentially the same results. This should be considered and discussed further.

Response:

The top-100 dataset is mainly used to explore the properties of treeLFA and carry out basic validation analyses. This dataset contains the most frequent diseases in UKB (the smallest prevalence is about 0.01, which corresponds to 4000 cases). For these very common diseases, the inference of multimorbidity patterns is easy, and the influence of the prior is minimal. This result is also consistent with the simulation results. In the discussion section, we further clarified this.

Changes: Manuscript line 491-494.

Comment 8

Lines 248-254 raise the issue that the topic-GWAS generally will have a bigger N and more power, and this is never addressed (as far as I can see) in the paper. It might be useful to disentangle the effect of bigger N for topics vs. their lack of specificity, in order to make more fair comparisons with the individual code-GWAS.

Response:

We have clarified in the text that the sample size for topic-GWAS and single code GWAS are the same.

We consider that the major goal of treeLFA is grouping individual diseases into multimorbidity clusters (topics) and increasing N (number of cases) for GWAS. For topic-GWAS, we are mainly interested in those loci having effects on the majority of diseases in a topic, in another word, those associated with a multimorbidity cluster instead of a specific disease. Technically it is quite difficult to disentangle the effects of bigger N and less specificity for some individual diseases for topic-GWAS.

As for the comparison of single code GWAS and topic-GWAS, making use of the test data and comparing the prediction ability of these two methods should be an objective method (Supplementary Figure 7A-B). We used the PRS for single codes as a metric and obtained interesting results.

Changes: Manuscript line 256.

Comment 9

I think the authors are missing an opportunity to highlight the main interesting findings; the discussion of the GWAS is fairly clear but does not address the key question directly: what are the SNPs that are topic-specific but NOT found in the code-GWAS (even after correction for improved power in the topic-GWAS)? These are the SNPs that will implicate genes and loci that might not be strong for individual diseases but are strong in the context of the topic. These are the main discoveries of interest and they are not sufficiently highlighted in this paper. Indeed, the validation strategy of looking at GWAS catalogue (e.g. line 308 ff) is counter-productive because we are interested in those SNPs that would NOT be in the GWAS catalogue because they are only "visible" in the context of the topic-GWAS and so this is a big lost opportunity that should be addressed in a revision.

Response

We thank the reviewer for the supportive comments and suggestions. In the discussion section the biological implications of topic-unique loci are further discussed.

For topic-associated loci, we used overlap with the GWAS catalogue as one approach to validate them as a whole. We agree with the reviewer that once we have arguments to trust the results, the novel associations are most interesting.

To make the result more convincing, we added another validation specific for those topic-associated SNPs not in the GWAS catalogue. For the empty topic, we also carried out more analyses to shed light on its biological significance.

Changes: Manuscript line 440-447, 534-538; Supplementary information line 170-186; Supplementary Figure 12.

Comment 10

Discussion around line 270 again is confused by the different power of topic-GWAS (more power) vs. code-GWAS (less power) and this must be addressed somehow in order for this discussion to make sense.

Response:

We clarified in the text that topic-GWAS and single code GWAS have the same sample size. We also clarified that in Figure 4C we compared the P-values of the same set of SNPs given by topic/single code-GWAS. This was to show that for certain SNPs topic-GWAS might have larger power due to the combination of relevant diseases into topics, therefore topic-GWAS can be a good supplement to single code GWAS.

Changes: Manuscript line 256, 273-275.

Comment 11

Discussion on lines 279-293 further makes me wonder if the claim of tree-based prior makes any sense or improves the performance in any meaningful way.

Response:

We consider that the direct use of internal codes (e.g. as traits for GWAS) is only a very basic way of using the tree-structured disease ontologies, therefore we shouldn't conclude that they cannot improve the topic modelling of diseases. The use of the tree prior is more flexible than using internal codes directly, since the assumption of the tree prior is that any disease codes that are close on the tree are likely to have similar probability in the same topic. This gives rise to large numbers of possible combinations of diseases as multimorbidity patterns.

In this paper, GWAS on internal codes as binary traits was mainly done to verify that topic-associated loci were not associated with the entire groups of diseases in the disease classification system, but more specific multimorbidity patterns.

Comment 12

The discussion on lines 316-323 is not convincing because those observations are true for virtually all GWAS hits that have ever been observed (as far as I can tell), and so this is a weak validation.

Response:

We agree that our functional validation should be true for all GWAS signals. The major aim of this secondary analysis is to validate the topic-GWAS result, since topic weights are inferred phenotypes, and topic-GWAS is a new method compared to single code GWAS. We have emphasised this in the text.

We have edited the text to summarise the results of these validation analyses. We agree that this is not a strong validation since we only compared the overall behaviours of three groups of SNPs. It remains possible that some associations for topics are false positives. Nevertheless, this analysis shows that large proportions of topic-associated loci should be true signals. Besides, the group of topic-associated SNPs analysed only contains those that cannot be found by single code GWAS, which means the effect of single code associations were excluded.

For loci associated with the empty topic and those loci not in the GWAS catalogue, we carried out additional analyses, and also obtained supportive results.

Changes: Manuscript line 336-339; Supplementary information line 170-186; Supplementary Figure 12.

Comment 13

The PRS calculations are useful and interesting, but again the AUCs should not be compared directly until something is done to normalise for the increased power of the topic-GWAS hits, etc...same issue as discussed above.

Response:

Here we build two types of PRS for the same traits – all the individual ICD-10 codes, one based on single code GWAS and the other based on topic-GWAS results. We consider that no normalisation should be needed here since we are comparing two ways of prediction evaluated using the same metric.

Reviewer #1 student

I shared this manuscript with a senior graduate student working in this area. I append their comments here, in case also useful to the authors or editors.

Thank you for the opportunity to review this paper. The authors identify comorbidity clusters - "topics" in their definitions - by treeLFA and a prior ICD-10 coding system. Then, they present the GWAS results on the topics compared to those of constituent disease codes. The authors aim to show how topic modelling helps discover multimorbidity patterns. Still, they need to provide more statistical justification on how the genetic analysis of the combined traits provides insights beyond the study of individual code based on their GWAS. Overall, the work is well-written and covers a great topic of genetics involved in multimorbidity pathogenesis that deserves more attention.

Comment 1

The use of ICD-10 could be error-prone. It concerns me that the authors have discussed little of the previous work related to ICD-10 and clinical applications of ICD-10. ICD-10 coding could be inaccurate and associated with issues such as missingness. These problems are likely connected to a frequently assigned "empty" topic that needs further discussion.

Response:

In the discussion section we have added limitations of this study, including the errors in assigning ICD-10 codes to individuals.

We did not explore the potential bias in assigning ICD-10 codes to individuals, since in this study we merely used the ICD-10 data as the input for our model. The main focus was put on developing a model to find multimorbidity patterns and the downstream analyses. In the future, it would be important and meaningful to find a way to take the reliability of diagnosed codes into account. Changes: Manuscript line 574-576.

Comment 2

Some common ICD-10 codes were not covered in Table 1, type 1 diabetes mellitus E10, for example. E11 (type 2 diabetes) was found in many disease topics, but no mention of E10 in any metabolic-related clusters. How is the number of 100 top codes decided to use instead of 120, 150, and 200? In addition, defining the disease clusters by parameter tuning (K) could be noisy. The optimal K is data size-dependent and involves more post-hoc selections if including more diagnostic codes.

Response:

In this study we mainly used the HES data in UKB, where E10 is not as common as many other diseases. As a result, E10 is not included in Table 1, where only top active codes in topics are shown. For our study the top-100 dataset is mainly used for methodological exploration, so the total number of diseases is arbitrarily chosen. By contrast, the top-436 dataset was chosen to explore the multimorbidity structure in the biobank, and is more biologically important since it includes many more diseases. The size of the top-436 dataset is close to the limit of our computational resources. For the clustering of topics, we have clarified in the Methods section that we used a relatively small value for k to ensure that small clusters will not be absorbed into larger ones. After the Louvain clustering, the hierarchical clustering on the averaged topic vectors of all clusters will further combine very similar topics. In reality, we found that usually only near-empty topics would remain after the first step of Louvain clustering. These topics are usually unstable and not biologically meaningful.

Changes: Manuscript line 960-962.

Comment 3

The authors leverage the ICD-10 architecture. ICD-10 structure is a classification of diagnostic codes which does not explicitly consider comorbidities. Can authors describe more about how the hierarchical structure of the ICD-10 codes could be helpful to them? What are the implications associated with good performance of using this outside information?

Response:

We have edited the text to clarify the effect and implications of the tree-prior. The ICD-10 system is one of the most widely used disease classification systems today, and the use of the tree-prior for topics is aimed to incorporate this information about the relationship of diseases into the modelling framework, as a supplement to the data-drive approach. The diagnosis data in UKB is very sparse and noisy, especially when a large number of diseases is modelled. The use of prior domain knowledge can help increase the stability and interpretability of the inference result. The good performance associated with this outside information on the top-436 UKB dataset showed that the prior knowledge does improve the generalisation ability of the inference result to the test data, though only to a limited extent.

Changes: Manuscript line 149-151, 494-498.

Comment 4

GWAS on topic weights makes sense but needs a more thorough analysis of why this could be useful to disease studies. The difference between the number of loci found by 11 topic-GWAS (82+46) and the single code GWAS (730) is significant and surprising, same big difference for the top-436 dataset. Can authors address why the topic-GWAS has only identified a small proportion of unique loci found

in the single code GWAS? How specifically does the relative prevalence of each disease code make an impact on the topic-GWAS? They need to adequately address these issues to conclude that topic-GWAS can provide suitable power for genetic discovery underlying the comorbidity patterns. Authors used the higher p-values of lead SNPs for certain diseases in the topic-GWAS to indicate their "satisfying" statistical power. This can be over-optimistic without addressing the low number of identified loci. Comparing the results to those of individual GWAS in depth would be more helpful. It is reported that 89% and 78.9% of unique loci and variants identified in their GWAS are in the GWAS catalogue. The authors discussed little of the remaining associations found in the topic-GWAS but not in the GWAS catalogue. Are those potentially true signals or noises? Also, the authors should note that the GWAS catalogue is incomprehensive and outdated. All the GWAS stats should be included in the supplementary.

Response:

The reviewer raised several concerns in this section, which we address in turn:

1)topic-GWAS identified only a small proportion of ICD-10 associated SNPs:

We consider that the main reason only a small proportion of single disease code associated loci were found by topic-GWAS is that the target loci of these two GWAS methods are in essence different, since topic-associated loci are in theory those associated with the majority of diseases in a multimorbidity cluster. If a locus is only associated with one but not the remaining top active diseases in a topic, then it is likely that topic-GWAS won't find it as significant. We added a simulation to further verify this.

2)There are topic-associated SNPs that are not in the GWAS catalogue:

For those topic-associated SNPs that are not in the GWAS catalogue, we did an additional validation using the same enrichment analyses, and overall the results showed clear enrichment consistent with the associations being true signals.

3)Other concerns:

The impact of individual diseases in a topic on topic-GWAS result should largely depend on the probability of these diseases in the topic.

The GWAS stats have been deposited in the Mendeley data and will be published alongside the paper (<https://data.mendeley.com/datasets/rft63p3jcd/draft?a=fd0d7bcd-2073-4cc0-a95b-46778a920c95>).

Changes: Manuscript line 280-285, 440-447, 1061-1073; Supplementary information line 44-47; Supplementary Figure 3C-D, Supplementary Figure 12;

Comment 5

Comparing topic-GWAS to the phecode-oriented GWAS needs to be clarified. Phecodes are inclusive and correspond to many ICD-10 codes that might not be present in the same "disease topic".

Response:

We have clarified in the text that the Phecode-GWAS is a secondary analysis for single code GWAS to assess the influence of using a different coding system on the GWAS results. We agree that making comparisons between topic-GWAS and Phecode-GWAS is not very strict, since the topics were learned from diagnosis data encoded using the ICD-10 system. A better way is to infer topics for Phecodes, but it might not be necessary to add more comparisons into the paper as Phecode-GWAS is only a secondary analysis.

Changes: manuscript line 256-258, 262-263.

Comment 6

For PRS, authors should report the statistical numbers on if topic-PRS is significantly higher than single-code PRS (t-test, etc. on all codes) for both top-100 and top-436 datasets. Can authors explain why they add individuals' PRS for the topics weighted by the probabilities of the ICD-10 codes of interest in each topic? How important is this?

Response:

We guess the reviewer wanted to see the test statistics for the topic-PRS versus single-trait PRS. Results of paired Wilcoxon signed rank test (for comparing PRS based on treeLFA and flatLFA) and two-proportion Z-test (for all PRS comparison results) have been added. The basic idea behind constructing PRS for ICD-10 codes using topic-GWAS result is to validate topic-GWAS result with the test data, and using single code GWAS as a reference. To construct PRS for ICD-10 codes using topic-GWAS result, a natural way is to use a weighted sum of PRS for topics (as introduced in the paper). Since many codes are active in more than one topic, this can be metaphorically thought of as decomposing the risk of a disease into different “pathways” in the form of disease topics. We also clarified this in the text. The significance of this result has also been added to the discussion section: firstly, it indicates that the topic-GWAS results are reliable, since they could be used to predict risks of many single diseases as good as or even better than that achieved by using single code GWAS results in the standard way; secondly, it suggests that for certain diseases, making use of the GWAS results for their comorbidity may improve the prediction performance. Changes: Manuscript line 348-349, 353, 476-477, 525-527.

Minor issues

Comment 1

The larger dataset usually refers to a higher number of individuals in the study. It should be more precisely stated (e.g., analyses on the top-100 and top-436 datasets).

Response:

The subheading has been changed to clarify the inclusion of more diseases. Changes: Manuscript line 391.

Comment 2

As briefly introduced by the authors, many other ways of extracting comorbidity patterns exist. A few more citations on those should be sufficient.

Response:

More citations for approaches to studying multimorbidity and other relevant topic models have been added to the introduction section. Changes: Manuscript line 68-70, 92-95.

Comment 3

How efficient is the treeLFA approach? How computationally expensive is it?

Response:

The computational resources used by treeLFA have been added to the Methods section. A computation cost figure has been added to the Analytic note. Changes: Manuscript line 760-764. Analytic note (line 188-208).

Reviewer #2

Authors present a topic model called treeLFA with 3 main differences in contrast to the well-known Latent Dirichlet Allocation (LDA) model: (1) to model binary code in the cross-section EHR data such as the HES in UK Biobank, Bernoulli likelihood was used instead of Categorical likelihood in LDA with conjugate prior changed from Dirichlet to Beta for ϕ ; (2) the index for the hyperparameters "a" for the topics-diseases Beta distribution $\phi \sim \text{Beta}(a_{i,0}, a_{i,1})$ are further parameterized by Bernoulli variable "i", which follows Markov process dictated by the transition probabilities ρ along the known ICD-10 taxonomy. Overall, the paper is clearly written with a good organisation. The technical details were also fully presented in the "Analytical notes" and easy to follow (given my expertise in topic modelling of administrative data such as the ICD codes). However, I still have quite a few comments.

Response:

We thank the reviewer for the detailed comments and suggestions on how to improve the paper. In the revised version, we made several major changes:

1) Relevant topic models for medical data:

We cited all the papers mentioned by the reviewer, and chose the graph embedded topic model (GETM) as the most relevant one, and included it into all the major comparative analyses in the paper.

2) Evaluation of topics:

We calculated topic coherence, topic diversity, and topics' concordance with ICD-10/Phencode groups to evaluate topics inferred by different topic models.

3) Empty topic:

We added several new analyses for the empty topic, including enrichment validation specifically done for the empty topic, tissue enrichment analysis, and association of the empty topic with other non-genetic factors. We also revised the discussion about the empty topic.

4) Scalability of treeLFA:

We added a computational cost figure to the Analytic note, and clarified the computational resources used by treeLFA.

5) Technical issues:

We clarified technical details regarding post-processing of inference results, topic-GWAS and PRS calculation. We used "Lassosum" to repeat the calculation of all PRS to make the results more convincing.

The remaining concerns were also addressed accordingly.

Major comments:

Comment 1

There have been several recently developed topic models for modelling EHR data with more efficient stochastic variational inference algorithms compared to the Gibbs sampling algorithm. For example, MixEHR infers multimorbidity by modelling several types of EHR data (Li et al., Nat Comm 2022). On the technical side, variational inference as implemented in the MixEHR will resolve the topic switching issues during the MCMC sampling since one only needs to take the converged topics at the end of the iteration. I suggest the authors compare treeLFA with MixEHR or at least cite the paper as a related method.

a. Li, Y. et al. Inferring multimodal latent topics from electronic health records. Nat Commun 11, 2536 (2020).

Response:

We have cited mixEHR in the introduction section as a relevant method with unique value and strength.

The main strength of MixEHR is analysing multimodal data, especially datasets that includes lab test results. In this study, only diagnosis data was used, and the main focus was put on finding multimorbidity patterns and the downstream analyses, instead of modelling very complex multimodal medical data. Therefore, we consider mixEHR is not the most suitable model for this study. If only diagnostic data is provided as input for mixEHR, it will have no significant difference with LDA.

Variational inference is a mainstream inference method for topic models and is being widely used. The MCMC and variational methods both have their advantages and disadvantages. We implemented LDA using a variational EM algorithm via the R-package “topicmodels”, and found the results (total number of associated loci and the PRS for single codes based on topic-GWAS results) were similar but slightly worse than that given by LDA using Gibbs sampling. Nevertheless, the comparison between MCMC and variational algorithms was not the focus of this study, therefore we didn't present this result in the paper. We clarified the advantage of using Gibbs sampling for the inference of treeLFA in the Methods section.

For treeLFA, variational inference is difficult due to the way we constructed the tree-prior for topics. In the future, this can be further explored to improve the computation efficiency of treeLFA.

Changes: Manuscript line 92-95, 756-760.

Comment 2

In terms of using disease taxonomy to model UKB data, a more recent approach called graph-embedded topic model uses node2vec to learn the embedding of the disease codes and then leverage that to infer topics (Wang et al 2022, 2023). Please mention this method as a related method or better yet compare it with your treeLFA.

a. Wang, Y., Benavides, R., Diatchenko, L., Grant, A. V. & Li, Y. A graph-embedded topic model enables characterization of diverse pain phenotypes among UK biobank individuals. *iScience* 25, 104390 (2022).

b. Wang, Y., Grant, A. V. & Li, Y. Implementation of a graph-embedded topic model for analysis of population-level electronic health records. *Star Protocol* 4, 101966 (2023).

Response:

The graph-embedded topic model (GETM) uses embeddings for diseases to incorporate prior domain knowledge, and is a splendid work. We cited it in the introduction section.

In terms of the basic model structure, GETM is more similar to LDA than treeLFA, since the occurrence of diseases (conditions) on individuals are modelled with Multinomial distributions instead of Bernoulli distributions. Therefore, we classified GETM as a LDA based topic model in the paper.

Following the reviewer's suggestion, we implemented the model and compared the inferred topics and topic-GWAS results to that given by the other three topic models. The results show that all the sparse disease topics inferred by treeLFA and LDA were also inferred by GETM, while the remaining denser topics were not. The topic-GWAS results show that treeLFA gave better prediction performance than GETM. Overall, we consider these results showing that compared to treeLFA, GETM gave results more similar to that given by LDA.

Changes: Manuscript line 96, 226-228, 238-245, 317, 364-365, 786-792, 978-982; Supplementary

information line 27-35, 66-67, 114-116; Supplementary Figure 2D-F, Supplementary Figure 4D, Supplementary Figure 7E.

Comment 3

When aggregating the frequency of ICD-10 codes over multiple visits, the UKB data can still have count data. In line 787, the author mentioned that they keep only one record if that happens. How frequent is this? This also shows the inadequacy of using Bernoulli as likelihood instead of binomial.

Response:

We showed the count data for diseases in Figure 1 in the Analytic note. The reason we did not use counts of diseases as input for topic models is that many diseases were repeatedly coded in the medical record as secondary reasons for hospitalisation. Therefore, the counts of recurrent codes are not informative.

Changes: Analytic note line 17-24.

Comment 4

A more technical issue is the computational expense: Bernoulli likelihood requires computing the entire ICD-10 code vocabulary for each subject whereas LDA only models the observed words. As the second premise in the introduction, the treeLFA model is set to tackle sparsity of ICD-10 codes. Yet the authors focused only on the top 100 or top 436 most frequent ICD-10 codes. Why not model *all* the ICD-10 codes and removing only the low frequent ICD-10 codes (e.g., the ones that occur in fewer than 5 subjects)? If scalability is the problem (as authors mentioned given the limited computing resource), please provide time complexity and clearly describe in a separate section called "The limitation of the study" as treeLFA does not scale to more than 450 ICD-10 codes. Indeed, the scalability issue is due to the fact that you model each code as Bernoulli, which force you to model *all* input ICD-10 codes for *each* subject as opposed to modelling only the *observed* ICD-10 codes for each subject as in the LDA.

Response:

At present scalability is the main weakness of treeLFA. With our current computational resources, it is not realistic to scale treeLFA up to more than a few hundreds disease codes for the whole UKB dataset. This has been clarified in the Methods section of the paper. In addition, a computational cost figure has been added to the Analytic note.

We used a threshold of 0.001 for prevalence to select ICD-10 codes for the top-436 dataset, which corresponds to about 500 cases in UKB. We reasoned that including more rare diseases may not be very necessary for now, since it can be hard to infer stable comorbidity patterns for them, and small numbers of cases also make the downstream association study difficult.

LDA has the advantage of being computationally efficient, yet modelling diseases using Bernoulli distributions makes the method sensitive to disease prevalence as well as disease patterns, and enables the discovery of the empty topic, which results in the independence of individuals' weights for other disease topics and brings additional power for the topic-GWAS. In the future, more efficient inference algorithms like variational Bayes may be explored for treeLFA.

Changes: Manuscript line 759-764. Analytic note line 188-208.

Comment 5

Line 233: "Individuals that were not diagnosed with any of the top-100 ICD-10 codes (629/2,000) have a weight near 1 for the empty topic". My question is why include those "empty documents" in the first place?

Response:

If the inference of disease topics is the only goal, then completely healthy individuals can be excluded. The only difference will be in the optimised hyperparameter alpha (Dirichlet prior for topic weights).

Nevertheless, in this study the major focus is put on topic-GWAS, therefore we need a way to include those completely healthy individuals into GWAS. We clarified this in the Discussion section. A near one weight for the empty topic and near zero weights for disease topics for healthy individuals is a reasonable result naturally given by treeLFA, and it is also a simple sanity check for the inference algorithm. Note that we find evidence for genetic determinants of the healthy topic, which - we feel - is an important finding of the research.

Changes: Manuscript line 541-554.

Comment 6

Line 255: besides showing the number of loci, what are the heritability estimates for those topics when treated as phenotypes?

Response:

The heritability of topic weights estimated using LDSC (LD score regression) with summary statistics of topic-GWAS are shown in Supplementary Table 10 for the top100 dataset and Supplementary Table 21 for the top-436 dataset.

Changes: Manuscript 431-432; Supplementary Table 10 and 21.

Comment 7

Line 260: how to explain that most unique loci were associated with the empty topic? I found this rather counter intuitive.

Response:

We are also intrigued by the findings relating to the empty topic. To investigate these further, we have carried out several additional analyses. First, we found evidence showing that the empty topic represents both an individual's disease burden and comorbidity structure. Second, we found that where there are variants associated with the empty topic in the GWAS catalogue, these are associated with lifestyle factors (Supplementary Figure 13C). The tissue enrichment analysis (sLDSC) also shows that the heritability of the empty topic is enriched in CNS cell types (Supplementary Figure 13D-E). These results suggest that the empty topic is closely related to an individual's multimorbidity burden and health behaviour.

Why should this topic be enriched for variants that are novel (compared to other topics)? We suggest that most GWAS have focused on disease traits where the controls are those without the disease in question. Some of these controls will have the healthy topic, but others will have different diseases. Variants that predispose to not having ANY disease (as opposed to a SPECIFIC disease) may therefore have been missed from previous analyses. We have updated the corresponding paragraph for the empty topic in the Discussion section to clarify these reasoning.

Changes: Manuscript line 448-464, 541-554; Supplementary information line 189-205, 301-302; Supplementary Figure 13; Supplementary Table 23.

Comment 8

Line 287: "insights" is an overstatement without biological support since several topics (other than the examples of topic 8 and topic 13) are not directly mapped to a single disease theme.

Response:

We agree with the reviewer that the word "insight" is an overstatement. We have rephrased the sentences to make it more objective.

As was mentioned above, for the empty topic we have done additional analyses to help unveil the underlying biological mechanisms. Yet in this paper, most of the other functional analyses were done to validate that the topic-associated loci were not noise, instead of revealing specific underlying mechanisms.

Changes: Manuscript line 300.

Comment 9

Please compare the predictive likelihood on the 20% test subjects among treeLFA, flatLFA, and LDA using both top-100 and top-436 UKB test data. I am aware that the authors have compared treeLFA with flatLFA in Supplementary Figure 10B as they described in line 295. However, this is an important experiment on real data and more direct compared to the AUC shown in Figure 4E (which is also valuable in its own light). I suggest doing a thorough comparison among the 3 methods on the

predictive likelihood and show them as a subpanel in Figure 4 in the main text. Also, please perform the comparison using the top-100 UKB data.

Response:

We have supplemented the comparison of predictive likelihood for treeLFA and flatLFA on the top100 dataset.

The predictive likelihood of treeLFA/flatLFA and LDA cannot be directly compared, as they model the same input data in different ways (as binary/count data using Bernoulli/Multinomial distributions). Their generalisation ability to the test data can be assessed in other ways, such as using prediction tasks on the test data. However, in this paper we want to focus on the association study using the inference result of topic models, since this is the major strength of treeLFA's model structure. This has been clarified in the text.

Changes: Manuscript line 231-232, 235-238; Supplementary information line 22-23; Supplementary Figure 2B.

Comment 10

Please also compute topic concordance to evaluate how the inferred topics phi concord with the expert-curated PheCodes or Clinical Classifications Software (CCS) codes: for each topic, use the highest topic coherence scores over all ICD groups to represent it; compare the distribution of those scores for flatLFA topics and those scores for LDA topics.

Response:

The Pearson correlation between topics inferred by the four topic models on the top-100 dataset and all ICD-10/Phecode groups (chapters/categories) were calculated. For each topic the top-5 ICD-10/Phecode groups which have the largest correlation with it were shown. The distribution of correlation between topics and the Phecode/ICD10 groups were plotted for all four topic models. In particular, GETM have larger correlation with ICD10/Phecode groups compared to the other three models, possibly due to the use of strong prior.

Changes: Manuscript line 240-245, 999-1008; Supplementary information line 30-35; Supplementary Figure 2E-F; Supplementary Table 7.

Comment 11

Line 416: The authors did not provide any interpretation of this seemingly rich figure. What are the biological implications of these enrichments? how to explain cases where the random loci give higher enrichment (e.g., chromHMM state 5 and 9)? Please elaborate on the findings of this figure.

Response:

A simple explanation of the enriched states and the full information of chromHMM states (Supplementary Table 11) have been added. Since each genomic locus will be in one chromHMM state, there will always be states (mostly functionally inactive states) for which the random loci are enriched compared to code- and topic-associated loci. Additional functional analyses have also been added, which show that larger proportions of topic-associated loci are in intronic and exonic regions on the genome compared to random loci. This is in accordance with the findings that topic-associated loci are likely to be in regions with strong transcription.

In general, the analysis of enrichment of chromHMM states among the three groups of loci was mainly intended to validate the topic-associated loci, by comparing its overall behaviours with random loci (as negative control) and single code associated loci (as positive control), instead of providing in-depth biological interpretations.

Changes: Manuscript line 325-332, 440-447; Supplementary Figure 12; Supplementary Table 11.

Comment 12

Line 427: For Supplementary Figure 10E, what's the p-value based, say Wilcoxon signed rank test or KS-test? They don't look significantly different.

Response:

The result given by paired Wilcoxon signed rank test has been added, which shows significant (P-value=0.029) difference between the PRS based on treeLFA/flatLFA results.

Changes: Manuscript line 476-477.

Comment 13

Line 468: PRSice-2 is often a baseline PRS method compared to LDpred2, Lassosum, SBayesR. Please show the PRS AUC using those methods to further home in this contribution.

Response:

We calculated PRS using the software Lassosum, and obtained very similar results as those given by PRSice-2, which verified that the PRS related results should be reliable.

Changes: Manuscript line 360-361, 1120-1121; Supplementary information line 106-107; Supplementary Figure 7B.

Comment 14

Line 653: The Analytical Notes are important details as integral parts of the manuscript. Given that this is a method paper, I suggest adding the model description to the Methods section in the main text. I also have quite a few comments on it as described below.

Response:

A more detailed introduction of treeLFA has been added to the beginning of the Methods section.

Changes: Manuscript line 718-753.

Comment 15

Line 700-702: This well written paragraph should be added to the model description as the rationale for *using* Markov process. The simulation just follows the same data generative process assumed by your treeLFA.

Response:

This has been added to the "treeLFA" subsection in the Methods section.

Changes: Manuscript line 728-742.

Comment 16

Supplementary Figure 9B does not look saturated to me.

Response:

We thank the reviewer for discerning this imprecise word. A more accurate description has been provided. For models set with a very large number of topics, some inferred topics remaining after clustering were very sparse and not stable, which caused the total number of topics and the predictive likelihood to fluctuate across different treeLFA models. Therefore, the plots didn't look perfectly saturated.

Changes: Manuscript line 384-387.

Comment 17

Line 717: in the simulation, are the first 3 topics the same or they just have the same Beta hyperparameters? Please clarify.

Response:

The first three topics are different as they have different active codes. All topics have the same beta priors for active/inactive disease codes. The descriptions of the simulated topics in the Methods section have been updated.

Changes: Manuscript line 811-812.

Comment 18

Line 741: Why not use the fixed point in Eq 7 to estimate alpha?

Response:

The simulation was intended to test if the tree prior for topics helps with the inference, so it was simplified as much as possible. Therefore, we didn't optimise the hyperparameter alpha in the simulation. This has been clarified in the Methods section.

Changes: Manuscript line 836-837.

Comment 19

The software page is incomplete. R markdown does not show up on the github browser.

Response:

The RMarkdown page for the demonstration of using treeLFA has been updated.

Changes: Manuscript line 1228-1229.

Comment 20

Topic identifiability. I invite authors to refer to MixEHR-Guide and MixEHR-Seed for solving the topic identifiability issue using an expert-curated guide (i.e., PheCodes).

a. Zhang, A. et al. Automatic Phenotyping by a Seed-guided Topic Model. Proc 28th Acm Sigkdd Conf Knowl Discov Data Min 4713-4723 (2022) doi:10.1145/3534678.3542675.

b. Ahuja, Y., Zou, Y., Verma, A., Buckeridge, D. & Li, Y. MixEHR-Guided: A guided multi-modal topic modelling approach for large-scale automatic phenotyping using the electronic health record. J Biomed Inform 134, 104190 (2022).

Response:

MixEHR-Guide and MixEHR-Seed are both extraordinary works aimed to infer biologically meaningful topics from high dimensional noisy data. They increase the interpretability of inferred topics by aligning them to important components or entities that are already known to us. In the Introduction section these two papers have been cited as relevant work.

The clustering method implemented in this study was aimed to solve a pure technical issue related to Gibbs sampling. In our study, we found that the accurate inference of topic weights for individuals was important to the downstream association analysis, and it was crucial to take account of the uncertainty of inferred topic weights by combining multiple posterior samples given by the Gibbs sampler, and this procedure significantly improved the topic-GWAS results. In summary, MixEHR-Guide and MixEHR-Seed and the clustering method in this paper deal with different research problems, therefore they may not be directly comparable.

Changes: Manuscript line 95-97.

Comment 21

Line 849: Why not use K-means with K set to be the same as the number of topics (i.e., $K=T$)? Also, some similarity threshold is needed since these clustering algorithms always form clusters regardless how similar the topics are in the same cluster.

Response:

K-means is a good clustering algorithm, yet the main point here is that we don't know the optimal number of topic clusters to keep. When a relatively large number of topics is set for the treeLFA model, there will be duplicated topics inferred, and we won't know the number of distinct topics.

Density based clustering algorithms like the Louvain algorithm also require specifying hyperparameters, but they have more flexibility compared to centroid based methods like K-means. The Methods section was updated and the motivations of using the Louvain algorithm was clarified. The similarity thresholds that can be used for the clustering is also supplemented. Changes: manuscript line 951-953, 976-977.

Comment 22

Line 857: Lots of post-hoc processing is done here, which undermines the original model elegance. It leaves the impression that the treeLFA model does not seem to work well and requires a lot of massaging at the end.

Response:

The post-processing of inference results is a part of the model selection strategy, and this has been clarified in the Methods section. Normally to find the optimal number of topics, we need to train many topic models with different numbers of topics, and then calculate their predictive likelihood or use other metrics to find the best one. For treeLFA, this is not computationally feasible on a very large dataset (such as the top-436 UKB dataset). Post-processing inference results is much easier than training multiple models, and for treeLFA it only involves collapsing and combining duplicated empty topics and near-empty topics. The description of this process in the Methods section is wordy, but this is fast and easy to implement.

For LDA, if a very large number of topics is set for the model, duplicated topics will also be inferred and need to be combined. Techniques like the Dirichlet process can automatically infer the optimal number of topics, but that is beyond the scope of this paper, and might be an interesting theme for future studies.

Changes: Manuscript line 965-966.

Comment 23

Line 873: While the logit ($\log(\theta/(1-\theta))$) makes sense, Dirichlet regression may fit better.

Response:

We agree that Dirichlet/Beta regression in theory are more suitable methods for modelling continuous proportions, but computationally they are much more expensive than linear regression combined with transformation of the response variable. Implementing them also involves solving technical issues like choosing the proper scale parameter for the regression. Transformation of the response variable is also a standard way to deal with situations where the assumptions of linear regression are violated. In future studies, more advanced regression techniques can be further explored.

Comment 24

Analytical Notes Line 40 for the choice of Beta priors: one benefit of LDA is to have a simplex Dirichlet over each topic such that the posterior topic distribution (also a Dirichlet due to conjugacy to the multinomial likelihood) will sum to 1 over the ICD-10 codes. This naturally yields diverse sets of topics. However, by having the Beta prior, each ICD-10 code under each topic sum to 1 over the active and inactive state. As a result, the topic distribution is much denser (since each topic does not sum to 1 over the ICD-10 codes). This may substantially decrease the topic interpretability and diminish the benefits of topic modelling. As a common quantitative metric to evaluate the topic quality, authors can evaluate topic diversity and topic coherence in comparison with LDA.

Response:

Topic coherence and topic diversity were calculated and added to the Result section. The coherence of topics inferred by treeLFA is slightly better than that given by LDA based topic models. The topic diversity of treeLFA topics is slightly worse than that for LDA based models. This can be caused by the fact that common diseases (such as I10(essential hypertension)) can have large probability in

multiple topics due to the lack of the constraint that probability of all codes in a topic sum to 1 for treeLFA, and topic diversity is only calculated using the top codes in topics.

Changes: Manuscript line 240-245, 983-998; Supplementary Table 6.

Comment 25

Qualitatively, I suggest displaying the top 5 active codes per topic for a select disease topics from their treeLFA and display their active state probabilities together in a heatmap. This is to show topic diversity and whether the top codes are meaningful under the topic (e.g., similar diseases should group together).

Response:

We have made interactive heatmaps for the 11 treeLFA topics on the top100 dataset and 40 treeLFA topics on the top-436 dataset as html files (supplementary files 1-2), where the top-10 active codes are annotated for all topics, and the meanings of all ICD-10 codes can be viewed in an interactive way.

Changes: Supplementary information line 363-377; Supplementary documents 2-3.

Comment 26

Can authors comment on the human bias in assigning the ICD-10 codes to each subject? For example, Song et al (2021) shows the benefits of using expert-specific topic distribution to model the expert domain knowledge.

a. Song, Z. et al. Supervised Multi-Specialist Topic Model with Applications on Large-Scale Electronic Health Record Data. in Proceedings of the 12th ACM Conference on Bioinformatics, Computational Biology, and Health Informatics 1-26 (Association for Computing Machinery, 2021). doi:10.1145/3459930.3469543.

Response:

We have cited this paper in the Introduction section as a relevant work. We agree with the reviewer that there will be bias in assigning ICD-10 codes to patients according to the judgments of humans, yet in this study we didn't deal with this issue. We have mentioned this as a limitation of this study in the Discussion section.

Changes: Manuscript line 574-576.

Minor comments

Comment 1

Line 214: should be "scale" instead of "normalise"

Response: This has been corrected as the reviewer suggested.

Changes: Manuscript line 203.

Comment 2

Line 297: "many fewer" => "much fewer"

Response: This has been corrected as the reviewer suggested.

Changes: Manuscript line 309.

Comment 3

Line 298-300: This does not seem to be a disadvantage of LDA to me since the empty topic is not useful anyway and the dense topic 8 is hard to interpret compared to the sparse topics. Negative correlation between topics imply topic diversity and having topic distribution sum to one over all codes imply sparsity. Neither is an unfavourable property of LDA.

Response:

As were mentioned before, we have added a few new analyses and discussions to show that the empty topic is also meaningful.

The negative correlation stated here is between individuals' weights for different topics, instead of between topics themselves. Due to the existence of the empty topic, individuals' weights for other disease topics are uncorrelated, which in theory should provide larger power for the downstream association study.

The main reason LDA topics are sparse is that we can directly control it by tuning the Dirichlet prior for topics. If we set $\eta=1$, the inferred topics will also be very dense. Like treeLFA, LDA also assign very small probabilities to most codes in a topic even if we set small values for η . When it comes to the interpretation, usually the top codes with large probability will receive the most attention.

As can be seen in Supplementary Figure 2, LDA also inferred dense topics that can be matched to treeLFA topics. Because LDA topics have the constraint that all codes' probability sum to 1, these topics didn't look as dense as corresponding treeLFA topics. Therefore, we consider the dense topics might also be meaningful in some way since they were inferred by different topic models.

Comment 4

Following my above recommendation, Figure 1 and 2 in the Analytical Notes should be added as subpanels to Figure 1 in the main text.

Response:

The graphical models have been added to Figure 1.

Changes: Manuscript line 619-621; Figure 1B-C.

Comment 5

Analytic Notes line 55-57: For the completeness, please add: $\rho_{10} = 1 - \rho_{11}$ and $\rho_{00} = 1 - \rho_{01}$ corresponding to from active-to-inactive and inactive-to-inactive transition.

Response:

The equations have been added to the Analytic note.

Changes: Analytic note line 64-66.

Comment 6

Please describe in the Appendix Figure 2 panel A legend that I_{ts} index the shape parameters "a" for Beta variable ϕ_{ts} (as shown in Equation 1 of the data generative process and Equation 3 for the conditional of ϕ_{ts}).

Response: This explanation has been added in the Analytic note.

Changes: Figure 3 in the Analytic note.

Comment 7

In Analytic Notes, Please add below Equation 3 that "Equation 3 was derived based on the fact that Beta is conjugate to Bernoulli likelihood."

Response: This has been corrected to make the derivation more complete.

Changes: Analytic note line 110.

Comment 8

In Analytic Notes line 137: "in topics" => "for all topics".

Response: This has been corrected.

Changes: Analytic note 135-142.

Comment 9

Lastly, the manuscript is long -- there are 78 PDF pages. While I don't think there is redundant content, it is difficult to navigate without hyperlink sections. Please add those to help the reviewing process.

Response: The final PDF file is generated by the submission system through combining all the individual files. As a result, bookmarks and hyperlinks won't work on the final PDF. I have added table of contents for the manuscript file, the point-to-point response file and the supplementary information file to help with the reviewing process.

Referees' report, second round of review

Reviewer 1

I thank the authors for their response to my reviews. They clarified some things and added a small number of new analysis. I think the paper is a good contribution.

Reviewer 2

All my comments were addressed. I would like to thank the authors for their efforts.

Authors' response to the second round of review

N/A